# E-Valuating Classifier Two-Sample Tests

**Teodora Pandeva**                                                    *t.p.pandeva@gmail.com*
*University of Amsterdam*

**Tim Bakker**                                                            *t.b.bakker@uva.nl*
*University of Amsterdam*

**Christian A. Naesseth**                                            *c.a.naesseth@uva.nl*
*University of Amsterdam*

**Patrick Forré**                                                        *p.d.forre@uva.nl*
*University of Amsterdam*

**Reviewed on OpenReview:** *https://openreview.net/forum?id=dwFRov8xhr*

## Abstract

We introduce a powerful deep classifier two-sample test for high-dimensional data based on E-values, called E-value Classifier Two-Sample Test (E-C2ST). Our test combines ideas from existing work on split likelihood ratio tests and predictive independence tests. The resulting E-values are suitable for anytime-valid sequential two-sample tests. This feature allows for more effective use of data in constructing test statistics. Through simulations and real data applications, we empirically demonstrate that E-C2ST achieves enhanced statistical power by partitioning datasets into multiple batches, beyond the conventional two-split (training and testing) approach of standard two-sample classifier tests. This strategy increases the power of the test, while keeping the type I error well below the desired significance level.

## 1 Introduction

We consider two-sample tests, which aim to answer the statistical question of whether two independently obtained populations are statistically significantly different. Often these tests help to distinguish real from generated data (Lopez-Paz and Oquab, 2017), noise from data (Hastie et al., 2001; Gutmann and Hyvärinen, 2012; Mikolov et al., 2013; Goodfellow et al., 2014), and are widely used in simulation-based inference (Lueckmann et al., 2021; Miller et al., 2022). In the general setting, consider the scenario where we are given two independent samples from two possibly different distributions:

$$X_1^{(0)}, \ldots, X_{N_0}^{(0)} \overset{\text{i.i.d.}}{\sim} P_0, \qquad\qquad X_1^{(1)}, \ldots, X_{N_1}^{(1)} \overset{\text{i.i.d.}}{\sim} P_1.$$

Based on these samples, we want to test if the distributions are equal or not. Thus, we can define a corresponding statistical test with null and alternative hypotheses as follows:

$$H_0: \ P_0 = P_1 \qquad \text{vs.} \qquad H_A: \ P_0 \neq P_1.$$

In this paper, we consider the two-sample testing problem in the context of sequential testing, where the user accumulates data from $P_0$ and $P_1$ in a time-dependent manner. The primary goal is to evaluate, at each observed time step, whether the null hypothesis defined above remains valid. Thus, if enough evidence against the null is acquired, we can reject the null and stop collecting data.

**Related work.** Two-sample testing has a long history in statistics, giving rise to classical techniques such as Student's and Welch's t-tests (Student, 1908; Welch, 1947), which compare the means of two normally distributed samples. In addition, nonparametric tests such as the Wilcoxon-Mann-Whitney test (Mann

and Whitney, 1947), the Kolmogorov-Smirnov tests (Kolmogorov, 1933; Smirnov, 1939), and the Kuiper test (Kuiper, 1960) have been established. In the area of high-dimensional data, kernel methods have been introduced, which focus on comparing the kernel embeddings of two populations (Gretton et al., 2012; Chwialkowski et al., 2015; Jitkrittum et al., 2016). However, these traditional two-sample statistical become less powerful when dealing with more complicated data types such as images and text.

Recent advances have led to the development of classifier-based two-sample tests (Kim et al., 2021) and their deep learning extensions, as seen in (Lopez-Paz and Oquab, 2017; Kirchler et al., 2020; Liu et al., 2020; Cheng and Cloninger, 2022). In these methods, a model is trained to discriminate between the two populations using training data, and then a statistical test is performed on a separate test set. However, all listed methods share a common limitation: when applied sequentially, they can lead to inflated type I error. In simpler terms, these methods assume that the sample size is known in advance, which can be a drawback in practice.

To address this limitation, sequential testing procedures offer a solution by allowing practitioners to dynamically reject the null hypothesis as new batches of data arrive. Within this context, E-value-based sequential tests have revived in the work of Shafer (2019); Ramdas et al. (2023); Grünwald et al. (2024), where they are interpreted as bets against the null hypothesis. More formally, E-variables are simply non-negative variables $E$ that satisfy

$$\text{for all } P \in \mathcal{H}_0 : \ \mathbb{E}_P[E] \leq 1,$$

i.e. the expectation of $E$ with respect to any distribution from the null hypothesis distribution class $\mathcal{H}_0$ is less than one. An example of E-variables for singleton hypothesis classes are Bayes factors, i.e. we test if the unknown probability density $p$ equals $p_0$ or $p_A$:

$$H_0 : p = p_0 \qquad \text{vs.} \qquad H_A : p = p_A.$$

The Bayes factor given by $E(x) := \frac{p_A(x)}{p_0(x)}$ is an E-variable w.r.t. $\mathcal{H}_0$ since $\mathbb{E}_{p_0}[E] = \int \frac{p_A(x)}{p_0(x)} p_0(x)\, dx = 1 \leq 1$. Note that observing a very large value of $E$, which we call an *E-value*, provides evidence against the null hypothesis.

The appealing theoretical properties of E−variables (details in Section 2.2) have led to the development of a growing body of work on E-value-based (conditional) independence tests spanning several domains, including two-sample tests for contingency tables (Turner and Grünwald, 2023), sequential data (Balsubramani and Ramdas, 2016), kernel-based approaches (Podkopaev et al., 2023; Shekhar and Ramdas, 2024), rank-based conditional independence tests (Duan et al., 2022), and conditional independence tests under 'model-X' assumptions (Grünwald et al., 2023; Shaer et al., 2023), split-likelihood two-sample tests (Lhéritier and Cazals, 2018), the concurrent work by Podkopaev and Ramdas (2024).

**Contributions.** We extend the work of Lhéritier and Cazals (2018) by introducing E-values for conditional independence testing which is a larger class of testing problems (see Section 4). The resulting E-values combine the ideas of the split-likelihood testing procedure of Wasserman et al. (2020) (see Section 3) and the existing work on predictive conditional independence testing frameworks of Burkart and Király (2017) (see Section 4). In contrast to (Lhéritier and Cazals, 2018), our framework, when applied to two-sample testing, referred to as E-C2ST, assumes a composite null hypothesis. Furthermore, it leverages the representational capabilities of machine learning models, allowing us to design tests for complex data structures.

We show that the described tests (including E-C2ST) provide non-asymptotic type I error control under the null, and are consistent (i.e., reject the null almost surely) in both a sequential and a non-sequential setting (details in Section 4). Moreover, when restricted to the two-sample test setting, we establish milder conditions required for the machine learning model than those in (Lhéritier and Cazals, 2018) for the consistency guarantees to hold.

In our empirical analysis in Section 6, we use the theoretical properties of E-C2ST to design sequential tests that optimize data usage by segmenting it into multiple batches. Each batch contributes to the cumulative test statistic. This method contrasts with traditional two-sample classifier tests, which derive a test statistic solely from the test set conditioned on the training data. Our approach not only achieves maximum power faster than standard methods, but also consistently keeps type I errors well below the significance level.

## 2 Hypothesis Testing with E-Variables

Building on the recent work of Ramdas et al. (2023); Grünwald et al. (2024), we give a detailed introduction to E-variables and their properties in Section 2.2 and establish their connections to hypothesis testing in Sections 2.3 and 2.4. We deferred all proofs to Appendix A. First, we introduce the notation used throughout this paper.

### 2.1 Notation

Consider a sample of data points $x_1, x_2, \ldots,$[1] reflecting realizations of random variables $X_1, X_2 \ldots$, drawn from an unknown probability distribution $P \in \mathcal{P}(\Omega)$ coming from some unknown sample space $\Omega$, where $\mathcal{P}(\Omega)$ is the set of all probability measures on $\Omega$. In hypothesis testing, we usually consider two model classes:

$$\mathcal{H}_0 = \{P_\theta \in \mathcal{P}(\Omega) \,|\, \theta \in \Theta_0\} \text{ (null hypothesis)},$$
$$\mathcal{H}_\mathsf{A} = \{P_\theta \in \mathcal{P}(\Omega) \,|\, \theta \in \Theta_\mathsf{A}\} \text{ (alternative)},$$

where $\Theta_0$ and $\Theta_\mathsf{A}$ represent the parameter sets of the distributions that are valid under the null and alternative, respectively. We want to decide if $P$ comes from $\mathcal{H}_0$ or from $\mathcal{H}_\mathsf{A}$:

$$H_0 : \ P \in \mathcal{H}_0 \qquad \text{vs.} \qquad H_\mathsf{A} : \ P \in \mathcal{H}_\mathsf{A}.$$

In most cases, the data points come from the same space $\mathcal{X}$ and we would at most observe countably many of such data points $X_n$. In this setting, we can w.l.o.g. assume that $\Omega = \mathcal{X}^\mathbb{N}$. If we, furthermore, assume that the $X_n$, $n \in \mathbb{N}$, are drawn i.i.d. from $P$ then $P((X_n)_{n\in\mathbb{N}}) = \bigotimes_{n=1}^\mathbb{N} P(X_n)$, we can directly incorporate the product structure into $\mathcal{H}_0$ and $\mathcal{H}_\mathsf{A}$ and restrict ourselves to one of those factors to state $\mathcal{H}_i$. By slight abuse of notations we re-write for $i = 0, \mathsf{A}$:

$$\mathcal{H}_i = \{P_\theta \in \mathcal{P}(\mathcal{X}) \,|\, \theta \in \Theta_i\},$$

and implicitly assume that $P_\theta((X_n)_{n\in\mathbb{N}}) = \bigotimes_{n=1}^\mathbb{N} P_\theta(X_n)$. Moreover, assume that our probability measures, $P_\theta \in \mathcal{H}_i$, are given via a density with respect to a product reference measure $\mu$. We denote the density by $p_\theta(x)$ or $p(x|\theta)$ interchangeably in this work.

### 2.2 Conditional E-Variables

Now consider the more general *relative* framework where we allow hypothesis classes to come from a set of Markov kernels, which can be used to model *conditional* probability distributions for $i = 0, \mathsf{A}$:

$$\mathcal{H}_i = \{P_\theta : \mathcal{Z} \to \mathcal{P}(\mathcal{X}) \,|\, \theta \in \Theta_i\} \subseteq \mathcal{P}(\mathcal{X})^\mathcal{Z}, \tag{1}$$

where $\mathcal{P}(\mathcal{X})^\mathcal{Z}$ denotes the space of all Markov kernels from $\mathcal{Z}$ to $\mathcal{X}$, i.e. for each $P_\theta \in \mathcal{H}_i$ for fixed $z \in \mathcal{Z}$ $P_\theta(\cdot|z)$ is a valid probability measure on $\mathcal{X}$. An example of conditional hypothesis classes is given in Section 4, where the null hypothesis class represent the set of distributions that reflect the conditional independence of two variables after observing a third one. With respect to $\mathcal{H}_0$ as defined in Equation (1) we can define corresponding E-variables which we call *conditional E-variables*[2][3]:

**Definition 2.1** (Conditional E-variable). *A conditional E-variable w.r.t. $\mathcal{H}_0 \subseteq \mathcal{P}(\mathcal{X})^\mathcal{Z}$ is a non-negative measurable map:*

$$E : \ \mathcal{X} \times \mathcal{Z} \to \mathbb{R}_{\geq 0}, \qquad (x, z) \mapsto E(x|z),$$

*such that for all $P_\theta \in \mathcal{H}_0$ and $z \in \mathcal{Z}$ we have:*

$$\mathbb{E}_\theta[E|z] := \int E(x|z) \, P_\theta(dx|z) \leq 1.$$

---

[1] In the following we will write small $x$ if we either mean the realization of a random variable $X$ or the argument of a function living on the same space. We use capital $X$ for a data point if we want to stress its role as a random variable.

[2] A formal definition of the "unconditional" E-variables introduced in Section 1 can be easily derived from Definition 2.1 by dropping $\mathcal{Z}$. Moreover, if $E$ is an E-variable and $x \in \mathcal{X}$ a fixed point then we call $E(x)$ the E-*value* of $x$ w.r.t. $E$.

[3] A similar definition can be found in (Grünwald et al., 2024).

One of the notable features of E-variables is their preservation under multiplication. We can easily combine (conditionally) independent E-variables by simply multiplying them which results in a proper E-variable. This property makes E-variables appealing for meta-analysis studies (Vovk and Wang, 2021; Grünwald et al., 2024). A more general result is that backward-dependent conditional E-variables can be combined by multiplication. This property of E-values is analogous to the chain rule observed in probability densities. Such a property becomes key in the development of the E-C2ST in this framework, which is formally stated as:

**Lemma 2.2** (Products of conditional E-variables (based on Grünwald et al. (2024))). *If $E^{(1)}$ is a a conditional E-variable w.r.t. $\mathcal{H}_0^{(1)} \subseteq \mathcal{P}(\mathcal{Y})^{\mathcal{Z}}$ and $E^{(2)}$ a conditional E-variable w.r.t. $\mathcal{H}_0^{(2)} \subseteq \mathcal{P}(\mathcal{X})^{\mathcal{Y} \times \mathcal{Z}}$ then $E^{(3)}$ defined via their product:*

$$E^{(3)}(x, y|z) := E^{(2)}(x|y, z) \cdot E^{(1)}(y|z),$$

*is a conditional E-variable w.r.t.:*

$$\mathcal{H}_0^{(3)} := \mathcal{H}_0^{(2)} \otimes \mathcal{H}_0^{(1)} \subseteq \mathcal{P}(\mathcal{X} \times \mathcal{Y})^{\mathcal{Z}},$$

*where we define the product hypothesis as:*

$$\mathcal{H}_0^{(2)} \otimes \mathcal{H}_0^{(1)} := \left\{ P_\theta \otimes P_\psi \;\middle|\; P_\theta \in \mathcal{H}_0^{(2)}, P_\psi \in \mathcal{H}_0^{(1)} \right\},$$

*with the product Markov kernels given by:*

$$(P_\theta \otimes P_\psi)(dx, dy|z) := P_\theta(dx|y, z) \, P_\psi(dy|z).$$

### 2.3 Hypothesis Testing with Conditional E-Variables

In the context of statistical testing, we can evaluate an E-variable based on the given data points, which are realizations of the random variables $X_1, \ldots, X_N$. Subsequently, the decision criterion for rejecting the null hypothesis at a significance level $\alpha \in [0, 1]$ is as follows

$$\textit{Reject } \mathcal{H}_0 \textit{ in favor of } \mathcal{H}_A \textit{ if } E(X_1, \ldots, X_N) \geq \alpha^{-1}.$$

Lemma 2.3 tells us that with this rule the type I error, the error rate of falsely rejecting the $\mathcal{H}_0$, is bounded by $\alpha$.

**Lemma 2.3** (Type I error control). *Let $E$ be a conditional E-variable w.r.t. $\mathcal{H}_0 \subseteq \mathcal{P}(\mathcal{X})^{\mathcal{Z}}$. Then for every $\alpha \in [0, 1]$, $P_\theta \in \mathcal{H}_0$ and $z \in \mathcal{Z}$ we have:*

$$P_\theta(E \geq \alpha^{-1}|z) \leq \alpha.$$

Thus, the E-values can be transformed into more conservative $p$-values via the relation $p = \min\{1, 1/E\}$ such that for $P_\theta \in \mathcal{H}_0$ it holds $P_\theta(p \leq \alpha|z) \leq \alpha$. Note that a valid way of constructing an E-variable from the independent random variables $X_1, \ldots, X_N$ w.r.t. the observed sample points according to Lemma 2.2 is $E(X_1, \ldots, X_N) = \prod_{i \leq N} E(X_i)$.

### 2.4 Sequential Hypothesis Testing with Conditional E-Variables

Up to this point, we have focused primarily on E-value-based tests for scenarios where the sample size $N$ is predetermined. Now suppose that the data does not arrive all at once, but instead we observe an infinite stream of data. In this context, a new sample $X_t$ becomes available at each time $t$. Consequently, we are interested in developing statistical tests that allow us to reject the null hypothesis at any given time $t$. These tests are called sequential tests and can be constructed using E-variables.

Building on the concepts introduced in the previous section, we can define a conditional variable $E^{(t)} = E(X_t|X_1, \ldots, X_{t-1})$, conditioned on past observations $X_1, \ldots, X_{t-1}$ with respect to the null hypothesis $\mathcal{H}_0^{(t)} \subseteq \mathcal{P}(\mathcal{X})^{\mathcal{X}^{t-1}}$. Importantly, Lemma 2.2 suggests combining all the evidence available up to time $t$ to construct a backward-dependent E-variable. In other words, the running product $E^{(\leq t)} = \prod_{l=1}^{t} E^{(l)}$, where $E^{(1)} = E(X_1)$, proves to be a valid E-variable with respect to $\mathcal{H}_0$.

This sequence of E-variables, also known as an E-process (Ramdas et al., 2023), offers a theoretical advantage over non-sequential $p$-value-based tests by allowing what it is called "optional continuation" (Grünwald et al., 2024). In simple terms, the user can make an informed decision at any given time $t$: whether to accumulate more data from additional experiments or to stop the process. This decision can be driven by, for example, the decision to reject the null hypothesis. The optional continuation property is facilitated by the following result, which ensures an anytime type-I-error bound for the process $(E^{(\leq t)})_{t \geq 1}$.

**Proposition 2.1** ((Ramdas et al., 2023; Grünwald et al., 2024)). *Let $E^{(\leq t)}$ be the running E-variable described above. Then for all $P_\theta \in \mathcal{H}_0$ and all $\alpha \in (0, 1]$*

$$P_\theta(\exists t \geq 0 \text{ such that } E^{(\leq t)} \geq \alpha^{-1}) \leq \alpha.$$

This result implies that we maintain type-I-error control not just at individual time points $t$ but consistently throughout the entire data collection period. More precisely, the decision rule for rejecting the null hypothesis:

*Reject $\mathcal{H}_0$ in favor of $\mathcal{H}_A$ if $E^{(\leq t)} \geq \alpha^{-1}$ for **any** $t \geq 1$*

has type I error bounded by $\alpha$. Additionally, we consider this sequential test to be consistent if it correctly rejects the null with a finite number of steps: $P_\theta(\exists t \geq 0 \text{ such that } E^{(\leq t)} \geq \alpha^{-1}) = 1$ for all $P_\theta \in \mathcal{H}_A$.

## 3 $M$-Split Likelihood Ratio Test

In general, constructing an E-variable with respect to any $\mathcal{H}_0$ is not a straightforward task. There exist two main approaches. The first approach, see (Grünwald et al., 2024), is based on the reverse information projection of the hypothesis space $\mathcal{H}_A$ onto $\mathcal{H}_0$. It is not data-dependent and can be shown to be growth-optimal in the worst case. However, the reverse information projection is not very explicit in general settings, especially when working with non-convex hypotheses, $\mathcal{H}_A$ and $\mathcal{H}_0$. The second approach is based on constructing a data-driven E-variable. By utilizing the $M$-split likelihood ratio test by Wasserman et al. (2020) introduced in this section, we establish an E-variable for a fixed sample size. Subsequently, we will demonstrate that the same E-variable can be adapted for sequential testing for an infinite data stream.

Assume that our data set $\mathcal{D} = \{X_1, \ldots, X_N\}$ is of size $N$. We now split the index set $[N] := \{1, \ldots, N\}$ into $M \geq 2$ disjoint batches: $[N] = \mathcal{I}^{(1)} \dot{\cup} \cdots \dot{\cup} \mathcal{I}^{(M)}$. For $m = 1, \ldots, M$ we also abbreviate:

$$\mathcal{I}^{(<m)} := \mathcal{I}^{(1)} \dot{\cup} \cdots \dot{\cup} \mathcal{I}^{(m-1)}, \qquad x^{(m)} := (x_n)_{n \in \mathcal{I}^{(m)}} \in \prod_{n \in \mathcal{I}^{(m)}} \mathcal{X} =: \mathcal{X}^{(m)}$$

and $x^{(<m)}$, $x^{(\leq m)}$, $\mathcal{I}^{(\leq m)}$, analogously. Then for $m = 2, \ldots, M$ we follow these steps:

1. Train a model on $\Theta_A$ on all previous points $x^{(<m)}$ in an arbitrary way (MLE, MAP, full Bayesian, etc.) and get $p_A(x|x^{(<m)})$. To achieve a high power of the test, the density $p_A(x|x^{(<m)})$ should reflect the true distribution in the best possible way to generalize well to unseen data.

2. Train a model on $\Theta_0$ on the data points of the current batch $x^{(m)}$ (conditioned on the previous ones $x^{(<m)}$) via maximum-likelihood fitting (MLE):

$$\hat{\theta}_0^{(\leq m)} := \hat{\theta}_0^{(m)}(x^{(\leq m)}) := \underset{\theta \in \Theta_0}{\operatorname{argmax}} \, p_\theta(x^{(m)}|x^{(<m)}),$$

and get: $p_0(x|x^{(\leq m)}) := p(x|x^{(<m)}, \hat{\theta}_0^{(m)}(x^{(\leq m)}))$. Note that under i.i.d. assumptions there is no dependence on $x^{(<m)}$.

3. Evaluate both models on the current points $x^{(m)}$ and define $E^{(m)}$ via their ratio:

$$E^{(m)}(x^{(m)}|x^{(<m)}) := \frac{p_A(x^{(m)}|x^{(<m)})}{p_0(x^{(m)}|x^{(\leq m)})} = \frac{p_A(x^{(m)}|x^{(<m)})}{\max_{\theta \in \Theta_0} p_\theta(x^{(m)}|x^{(<m)})}. \tag{2}$$

Then $E^{(m)}$ constitutes a conditional E-variable, conditioned on the space $\mathcal{X}^{(<m)}$, w.r.t. $\mathcal{H}_0^{(m)|(<m)}$ (see Appendix A for the proof).

From the previous section and Lemma 2.2 we know that the running product

$$E := E^{(\leq M)} := \prod_{m=1}^{M} E^{(m)},$$ (3)

defines an E-variable w.r.t. $\mathcal{H}_0 = \mathcal{H}_0^{(\leq M)}$ (see Appendix A for the proof). For fixed $M$, Lemma 2.3 gives us type I guarantees of the derived test. In other words, the $M$-split likelihood ratio test, for significance level $\alpha \in [0,1]$, rejects the null hypothesis $\mathcal{H}_0$ if $E(X_1, \ldots, X_N) \geq \alpha^{-1}$ with type I error bounded by $\alpha$.

**Remark 3.1** (Intuition for the $m$-th conditional E-variable.). *For a fixed $m$, the $m$-th conditional E-variable $E^{(m)}$ intrinsically **compares the $\mathcal{H}_A$-model's test performance:** $-\log p_A(x^{(m)}|x^{(<m)})$, which is trained on $x^{(<m)}$, tested on $x^{(m)}$, **with the $\mathcal{H}_0$-model's train performance:** $-\log p_0(x^{(m)}|x^{(m)})$, both trained and tested on the same $x^{(m)}$ in the i.i.d. case. This means that if the alternative is true, then the $\mathcal{H}_A$-model $p_A$ has to perform better on $x^{(m)}$ than the $\mathcal{H}_0$-model $p_0$, while the latter was allowed to directly be (over)fitted on $x^{(m)}$.*

Now consider a scenario where the number of batches $M$ is not fixed, and we are dealing with a continuous stream of incoming batches of data. Using the insights from Proposition 2.1, the resulting E-value from (3) can be used for sequential testing. In this way, we can achieve an even more robust form of batch-wise anytime type I error control (details in Appendix A for the proof), as follows:

**Corollary 3.2** (Batch-wise anytime type I error control). *Consider the sequence of E-variables $\left(E^{(\leq M)}\right)_{M \in \mathbb{N}}$ from equation Equation (3) for an infinite stream of finite batches of data points. It follows that for every $P_\theta \in \mathcal{H}_0$ and every $\alpha \in (0,1]$ we have the anytime type I error bound:*

$$P_\theta \left( \exists M \in \mathbb{N} \; E^{(\leq M)} \geq \alpha^{-1} \right) \leq \alpha.$$

### 3.1 Test Consistency

In this section, we explore the consistency of the test introduced earlier, both for the specific case of $M = 2$ and for that of $M = \infty$, addressing standard predictive testing and sequential testing scenarios. Consistency is a key concept in statistical testing, as it guarantees that as more data are accumulated under the alternative, the test becomes more reliable in accurately detecting true hypotheses. This property allows us to examine how the test behaves as the sample size increases.

Now consider the split-likelihood case for $M = 2$ and the singleton set $\mathcal{H}_0 = \{P_0\}$ with a density $p_0$ for which the E$-$variable is given by

$$E^{(N^{(2)}|N^{(1)})}(x^{(2)}|x^{(1)}) := E^{(1)}(x^{(2)}|x^{(1)}) = \prod_{n \in \mathcal{I}^{(2)}} \frac{p_A(x_n|x^{(1)})}{p_0(x_n)},$$

where with $E^{(N^{(2)}|N^{(1)})}$ we want to make explicit the dependence of the E$-$variable on the train $(N^{(1)})$ and test $(N^{(2)})$ data sizes. By making mild assumptions about the learner and ensuring the boundedness of the (conditional) E-variable, we prove (details in Appendix B.1) the consistency of the test with respect to the E-variable defined above:

**Theorem 3.3.** *Let $\mathcal{H}_0 = \{P_0\}$ be a singleton set. Consider a model class $\mathcal{H}_A$ and a learning algorithm that for every realization $x = (x_n)_{n \in \mathbb{N}} \in \mathcal{X}^{\mathbb{N}}$ and every number $N^{(1)} \in \mathbb{N}$ fits a model $P_A^{|x^{(1)}} \in \mathcal{P}(\mathcal{X})$ to the first $N^{(1)}$ entries $x^{(1)} = (x_n)_{n \in \mathcal{I}^{(1)}}$ of $x$. Assume that for every $P_\theta \in \mathcal{H}_A$ and $P_\theta$-almost every i.i.d. realization $x = (x_n)_{n \in \mathbb{N}}$ of $P_\theta$ there exists a number $N^{(1)}(x) \in \mathbb{N}$ such that for all $N^{(1)} \geq N^{(1)}(x)$:*

$$\mathrm{KL}(P_\theta \| P_A^{|x^{(1)}}) < \mathrm{KL}(P_\theta \| P_0), \qquad \sup_{x_n \in \mathcal{X}} |\log E(x_n|x^{(1)})| < \infty.$$ (4)

*Then, $P_\theta \left( E^{(N^{(2)}|N^{(1)})} < \alpha^{-1} \right) \to 0$ for $N^{(1)}, N^{(2)} \to \infty$.*

To ensure the consistency of the test, certain conditions are required. The first key assumption is that, given a sufficiently large data set, the output model $P_{\mathsf{A}}^{|x^{(1)}}$ has a greater fit to the class of alternative hypotheses than the null distribution has to the same class. The second assumption implies that both $p_{\mathsf{A}}^{|x^{(1)}}$ and $p_0$ must have lower bounds. In other words, it requires that $\inf_{x \in \mathcal{X}} p_{\mathsf{A}}^{|x^{(1)}}(x) > 0$ and $\inf_{x \in \mathcal{X}} p_0(x) > 0$. For example, this condition can be easily satisfied for certain discrete distributions, such as the categorical distribution, as shown in Section 5. A more precise formulation of this theorem can be found in Theorem B.6. Lhéritier and Cazals (2018) discuss similar but stronger assumptions. In their work, $P_{\mathsf{A}}^{|x^{(1)}}$ is required to be strongly pointwise consistent, i.e. $P_{\mathsf{A}}^{|x^{(1)}}(x) \overset{N \to \infty}{\longrightarrow} P_\theta(x)$ almost surely, for which they can provide $\lambda$-consistency results (a weaker notion of consistency). They also assume that the null hypothesis is known. Next, we will show that in the case $M = \infty$, we can remove this condition and only assume that the learner is a better approximation of the true distribution than the estimated null one. Under similar conditions as in Theorem 3.3 we can prove that the sequential test defined in Equation (3) is consistent. The proof is deferred to Appendix B.2.

**Theorem 3.4.** *Consider the sequence of E-variables $\left(E^{(\leq M)}\right)_{M \in \mathbb{N}}$ from Equation (3) for an infinite stream of finite batches of data points. Let the learning algorithm fit a model $P_{\mathsf{A}}^{|x^{(<M)}} \in \mathcal{P}(\mathcal{Z})$ to the first $M-1$ batches $x^{(<M)} = (x_n)_{n \in \mathcal{I}^{(<M)}}$. Assume that for every $P_\theta \in \mathcal{H}_{\mathsf{A}}$ and for all $M \in \mathbb{N}$ and every instantiation of $x^{(\leq M)}$ the learner satisfies*

$$\mathbb{E}_\theta \left[\log \frac{p_\theta(x^{(M)})}{p_0(x^{(M)}|x^{(M)})}\right] - \mathrm{KL}\left(P_\theta^{x^{(M)}} \| P_{\mathsf{A}}^{x^{(M)}|x^{(<M)}}\right) > r_M > 0, \qquad \sup_{x \in \mathcal{X}^{|\mathcal{I}^{(M)}|}} |\log E(x|x^{(<M)})| < s_M,$$

*where for the positive sequences $(r_M)_{M \in \mathbb{N}}$ and $(s_M)_{M \in \mathbb{N}}$ hold*

$$\limsup_{M \to \infty} \frac{1}{M} \sum_{m=1}^{M} r_m > 0, \qquad\qquad \lim_{M \to \infty} \sum_{m=1}^{M} \frac{s_m^2}{m^2} < \infty$$

*Then, $P_\theta(\exists M \in \mathbb{N} \text{ such that } E^{(\leq M)} \geq \alpha^{-1}) = 1$.*

The requirement regarding the first sequence can be understood as guaranteeing that the learner consistently provides a better approximation of the true distribution compared to the estimated null distribution, averaged over a sequence of $M$ consecutive steps. In this context, $r_M$ could even be a decreasing null sequence, such as $\log(M)/M$. The second condition is a milder assumption than uniform boundedness. Similar to the previous result, this condition is relatively easier to satisfy when dealing with categorical random variables or random variables with a compact support.

## 4 Predictive Conditional Independence Testing

In this section, we combine the ideas of predictive conditional independence testing by Burkart and Király (2017) with E-variables from the $M$-split likelihood ratio test from Section 3 based on Wasserman et al. (2020) to derive a proper E-variable for *conditional* independence testing. The desired two sample test will later on be reformulated as an independence test by utilizing the theoretical results discussed in this section.

As a reminder, in conditional independence testing, we want to test if a random variable $X$ is independent of $Y$, or not, conditioned on $Z$:

$$H_0: \ X \perp\!\!\!\perp Y \,|\, Z \qquad \text{vs.} \qquad H_{\mathsf{A}}: \ X \not\!\perp\!\!\!\perp Y \,|\, Z,$$

based on data $\mathcal{D} = \{(X_1, Y_1, Z_1), \ldots, (X_N, Y_N, Z_N)\}$. The corresponding (*full*) hypothesis spaces, in the i.i.d. setting, are:

$$\mathcal{H}_0^{\mathrm{fl}} = \{P_\theta(X|Z) \otimes P_\theta(Y|Z) \otimes P_\theta(Z) \,|\, \theta \in \Theta_0\} \qquad \mathcal{H}_{\mathsf{A}}^{\mathrm{fl}} = \{P_\theta(X, Y, Z) \,|\, \theta \in \Theta_{\mathsf{A}}\} \setminus \mathcal{H}_0.$$

If we assume that $P(X, Z)$ is *fixed* for $\mathcal{H}_0$ and $\mathcal{H}_{\mathsf{A}}$ then this simplifies to the following product hypothesis classes, $i = 0, \mathsf{A}$: $\mathcal{H}_i^{\mathrm{fx}} := \mathcal{H}_i^{\mathrm{pd}} \otimes \{P(X, Z)\}$, where the conditional hypothesis classes $\mathcal{H}_i^{\mathrm{pd}} \subseteq \mathcal{P}(\mathcal{Y})^{\mathcal{X} \times \mathcal{Z}}$ of *predictive distributions* are given by:

$$\mathcal{H}_0^{\mathrm{pd}} = \{P_\theta(Y|Z) \,|\, \theta \in \Theta_0\}, \qquad\qquad \mathcal{H}_{\mathsf{A}}^{\mathrm{pd}} = \{P_\theta(Y|X, Z) \,|\, \theta \in \Theta_{\mathsf{A}}\} \setminus \mathcal{H}_0. \qquad (5)$$

Equation (2) applied to $\mathcal{H}_i^{\mathrm{fx}}$ under i.i.d. assumptions leads us to the following $m$-th conditional E-variable, using the abbreviation $w = (x, y, z)$:

$$E^{(m)}(w^{(m)}|w^{(<m)}) = \frac{p_{\mathsf{A}}(y^{(m)}|x^{(m)}, z^{(m)}, w^{(<m)})}{p(y^{(m)}|z^{(m)}, \hat{\theta}_0(y^{(m)}, z^{(m)}))}. \tag{6}$$

The reason that we applied Equation (2) to $\mathcal{H}_i^{\mathrm{fx}}$ instead of $\mathcal{H}_i^{\mathrm{fl}}$ is that $E^{(m)}$ will automatically be a valid conditional E-variable for $\mathcal{H}_0^{\mathrm{fl}}$ and even $\mathcal{H}_0^{\mathrm{pd}}$, as well.

**Remark 4.1.** *According to Shah and Peters (2020), in the general case, a valid test for conditional independence lacks power against all alternatives unless certain additional assumptions are introduced. One such assumption is the model $X$ assumption, where the user can access the conditional distribution $P(X|Z)$. In the construction outlined above, this assumption is implicit, since we assume that the joint distribution $P(X, Z)$ remains fixed under both hypotheses.*

## 5 Classifier Two-Sample Tests with E-Variables

---
**Algorithm 1** Algorithmic description of E-C2ST.

---
1: **Input:**
     Data stream $(x^{(m)}, y^{(m)} = (x_n, y_n)_{n \in \mathcal{I}^{(m)}})_{m \in [M]}$, Significance level $\alpha$, Initial $\lambda_1$,
     Training epochs $T$
2: **Initialize:**
     $\mathcal{D}_{train}, \mathcal{D}_{val} \leftarrow \mathrm{Split}((x_n, y_n)_{n \in \mathcal{I}^{(1)}})$, $E^{(1)} = 1$
3: **for** $m = 2, \ldots, M$ **do**
4:     **for** $t = 1, \ldots T$ **do**
5:         $g_\theta \leftarrow \mathrm{Train}(g_\theta, \mathcal{D}_{train})$
6:         **if** $\mathrm{EarlyStopping}(g_\theta, \mathcal{D}_{val})$ **then**
7:             **break**
8:     Compute $E^{(m)}$ on $(x_n, y_n)_{n \in \mathcal{I}^{(m)}}$ as in Equation (9).
9:     Compute $E := \prod_{m=1}^{M} E^{(m)}$
10:     **if** $E > \alpha^{-1}$ **then**
11:         **reject** and **break**
12:     Obtain $\lambda_{m+1}$ that solves (10)
13:     $\mathcal{D}_{train} \leftarrow \mathcal{D}_{train} \cup (x_n, y_n)_{n \in \mathcal{I}^{(m-1)}}$, $\mathcal{D}_{val} \leftarrow (x_n, y_n)_{n \in \mathcal{I}^{(m)}}$

---

In this section, we will formalize the classifier two-sample test using E-variables. The following test can be easily derived from the conditional independence test introduced in Section 4 by introducing a binary variable denoted $Y$, along with the following abbreviation:

$$P(X|Y=0) := P_0(X), \qquad\qquad P(X|Y=1) := P_1(X),$$

If we pool the data points $X_n^{(y)}$ via augmenting them with a $Y$-component: $(X_n^{(y)}, Y_n^{(y)})$ with $Y_n^{(y)} := y$, then the pooled data set can be seen as one i.i.d. sample from $P(X, Y)$ of size $N := N_0 + N_1$ for some unknown marginal $P(Y)$. We can then reformulate the two-sample test as an independence test:

$$H_0 : \ X \perp\!\!\!\perp Y \qquad \text{vs.} \qquad H_{\mathsf{A}} : \ X \not\!\perp\!\!\!\perp Y.$$

This allows us to use the E-variables from Section 4 (without any conditioning variable $Z$) for (conditional) independence testing. Furthermore, since $Y$ is a binary variable we can write any Markov kernel $P(Y|X)$ as a Bernoulli distribution $P_\theta(Y|X=x) = \mathrm{Ber}(\sigma(g_\theta(x)))$ for some parameterized measurable function $g_\theta$ and where $\sigma(t) := \frac{1}{1+\exp(-t)}$ is the logistic sigmoid-function. So our hypothesis spaces look like:

$$\mathcal{H}_0 = \{\mathrm{Ber}(q_\theta) \,|\, \theta \in \Theta_0\}, \qquad\qquad q_\theta \in [0, 1], \tag{7}$$
$$\mathcal{H}_{\mathsf{A}} = \{\mathrm{Ber}(\sigma(g_\theta)) \,|\, \theta \in \Theta_{\mathsf{A}}\} \setminus \mathcal{H}_0,$$

and the $m$-th conditional E-variable is given by:

$$E^{(m)}(y^{(m)}|x^{(m)}, x^{(<m)}, y^{(<m)}) = \frac{p_{\mathsf{A}}(y^{(m)}|x^{(m)}, x^{(<m)}, y^{(<m)})}{p(y^{(m)}|\hat{\theta}_0(y^{(m)}))} \tag{8}$$

$$= \prod_{n\in\mathcal{I}^{(m)}} \left(\frac{\sigma(g_{\hat{\theta}_{\mathsf{A}}^{(<m)}}(x_n))}{N_1^{(m)}/N^{(m)}}\right)^{y_n} \cdot \left(\frac{1 - \sigma(g_{\hat{\theta}_{\mathsf{A}}^{(<m)}}(x_n))}{N_0^{(m)}/N^{(m)}}\right)^{1-y_n} .$$

The maximum likelihood estimator for $q_\theta$ with respect to $y^{(m)}$ is represented by $\hat{q}^{(m)} = N_1^{(m)}/N^{(m)}$. This estimate corresponds to the frequency of data points in the $m$-th batch that are classified as belonging to class $y = 1$. Additionally, the function $g_\theta$ is trained on the data set $(x^{(<m)}, y^{(<m)})$ using binary classification.

Therefore, we combine all conditional E-variables to create an anytime valid E-variable, as explained in Section 3. In addition, we can use Theorem 3.4 to prove the consistency of the classifier's two-sample test. Consequently, we introduce a minor adjustment to the conditional E-variable in (8), resulting in a bounded E-variable that yields the following consistency result:

**Lemma 5.1.** *Let the learning algorithm fit a model $P_{\mathsf{A}}^{|x^{(<m)}, y^{(<m)}} \in \mathcal{P}(\mathcal{Z})$ to the first $m-1$ batches $(x^{(<m)}, y^{(<m)}) = (x_n, y_n)_{n\in\mathcal{I}^{(<m)}}$. Furthermore, let $\tilde{P}_{\mathsf{A}}^{|x^{(<m)}, y^{(<m)}} := \lambda_m P_0^{|y^{(m)}} + (1-\lambda_m)P_{\mathsf{A}}^{|x^{(<m)}, y^{(<m)}}$ with corresponding density $\tilde{p}_{\mathsf{A}}^{|x^{(<m)}}$ and $\lambda_m \in (0,1)$. Then*

$$\tilde{E}^{(m)}(y^{(m)}|x^{(m)}, x^{(<m)}, y^{(<m)}) = \frac{\tilde{p}_{\mathsf{A}}(y^{(m)}|x^{(m)}, x^{(<m)}, y^{(<m)})}{p(y^{(m)}|\hat{\theta}_0(y^{(m)}))} \tag{9}$$

*constitutes a bounded conditional E-variable w.r.t (7), i.e. for every instantiation $(x^{(<m)}, y^{(<m)})$ it holds $\sup_{(x,y)\in\mathcal{X}\times\mathcal{Y}} |\log E^{(m)}(y|x, x^{(<m)}, y^{(<m)})| < \infty$. Additionally, if the batch sample size is at most $B \in \mathbb{N}$ together with rest of the conditions in Theorem 3.4 it follows that the E-variable $(E^{(\leq M)})_{M\in\mathbb{N}}$ with increments defined in (9) yields a consistent test.*

In the above lemma, we also make the assumption that the batch size cannot exceed a fixed number $B \in \mathbb{N}$. This assumption is not particularly restrictive, since we have the flexibility to construct the conditional E variable using a maximum of $B$ samples from each incoming batch. Furthermore, one can think of the sequence $\lambda_m$ as a sequence of hyperparameters. We propose a hyperparameter selection procedure that derives $\lambda_m$ from the previous step, leading to the constrained optimization problem:

$$\lambda_{m+1} \in \underset{\lambda\in(0,1)}{\operatorname{argmax}} \log \tilde{E}^{(m)} \equiv \underset{\lambda\in(0,1)}{\operatorname{argmax}} \sum_{n\in\mathcal{I}^{(m)}} \log\left(\frac{\lambda \cdot p(y_n|\hat{\theta}_0(y_n)) + (1-\lambda) \cdot p_{\mathsf{A}}(y_n|x_n, x^{(<m)}, y^{(<m)})}{p(y_n|\hat{\theta}_0(y_n))}\right). \tag{10}$$

In step $m+1$, we determine the mixing parameter $\lambda_{m+1}$ to optimize the conditional E value calculated in the previous step $m$, as given in (9). When considering an alternative hypothesis, we expect a general decrease in $\lambda_m$ with each successive step due to the model's enhancement with the accumulation of more data. Under the null hypothesis, this parameter is expected to maintain a higher value. Thus, this mixing parameter becomes important in balancing our assumptions about the most plausible hypothesis class. The resulting test we call E-C2ST whose steps are summarized in Algorithm 1.

## 6 Experiments

We compare our method to other classifier two-sample tests on the Blob, MNIST, and KDEF data. We will empirically show that we can exploit the ability of E-C2ST to construct test statistics by using the entire dataset to gain statistical power over the other tests that compute a $p$-value based only on the train-test data split. Meanwhile, E-C2ST keeps the type I error strictly below the alpha significance level.

### 6.1 Baselines Implementation and Training

We compare E-C2ST to the following baselines: **S-C2ST** (standard C2ST), is the C2ST proposed by Lopez-Paz and Oquab (2017); Kim et al. (2021); **L-C2ST** (logits C2ST) by Cheng and Cloninger (2022), and

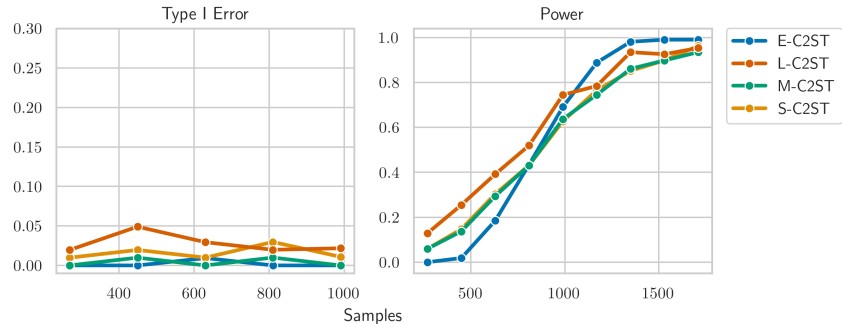

Figure 1: Type I error and Power experiments for the Blob dataset. Compared to the baselines, E-C2ST reaches maximum power faster than the baselines while maintaining the type I error strictly below the significance level.

**M-C2ST** by Kirchler et al. (2020). Each method involves training a classifier to differentiate the two classes and then using the trained model for computing the test statistics. The training procedure is with early stopping, and we use the same fixed network architecture across all tests for all experiments. The resulting $p$-values are reported from 500 permutations for all baseline tests. The main paper or the appendix details all experiments and tests' implementations (see Appendix D).

## 6.2 Evaluation

For each sample size we sample a dataset from a fixed distribution or as a subsample from a given data set. In the baseline case, we split the dataset into train, validation and test sets with ratio 5:1:1 and we fit a classifier. Using a significance level of $\alpha = 0.05$, we decide whether to reject the null hypothesis on the test set. In comparison, E-C2ST is evaluated sequentially. More precisely, for each fixed sample size, we sample a data set that is divided into batches of equal size. We apply the procedure described in Section 5 by constructing the running E variable as in (3) with initial $\lambda_1 = 0.5$ in all experiments unless specified otherwise. We decide whether to reject the null or to continue testing each time a new batch is observed. We run 100 independent experiments and report the rejection rates for all methods, corresponding to type I error or power, depending on whether the two classes are from the same distribution or not.

## 6.3 Empirical Results

**Blob Dataset.** The blob data set is a two-dimensional Gaussian mixture model with nine modes arranged on a $3 \times 3$ grid, used by Gretton et al. (2012); Chwialkowski et al. (2015) in their analysis. The two distributions differ in their variance, as visualized in Figure 6 in Appendix D.2. In the case of E-C2ST, we split the data into mini-batches of size 90 and compute the test statistics as explained in the previous section. The results are plotted in Figure 1, where the x-axis refers to the sample sizes of the total data used and the y-axis is the rejection rate. E-C2ST reaches maximum power faster than the baseline methods while achieving type I error strictly below the $\alpha$ significance level.

**KDEF Data.** The Karolinska Directed Emotional Faces (**KDEF**) dataset (Lundqvist et al., 1998) is used by Jitkrittum et al. (2016); Lopez-Paz and Oquab (2017); Kirchler et al. (2020) to distinguish between positive (happy, neutral, surprised) and negative (afraid, angry, disgusted) emotions from faces. We draw datasets of sizes $3 \cdot 64, 4 \cdot 64, \ldots$ from the two classes on which we perform the statistical tests. We set the E-C2ST batch size to 64 and run 100 independent experiments per sample size. The results are shown in Figure 2, where we compute the rejection rate (type I error and power) per sample size. Although our method has lower power for smaller sample sizes than the baselines, it reaches maximum power faster while having a type I error consistently below 0.05.

**Corrupted MNIST Data.** The MNIST dataset (LeCun et al., 1998) consists of 70 000 handwritten digits. As in (Liu et al., 2020), we make use of generated MNIST images (LeCun et al., 1998) from a pre-trained DCGAN model (Radford et al., 2015) for this benchmark experiment. More precisely, we assume that under

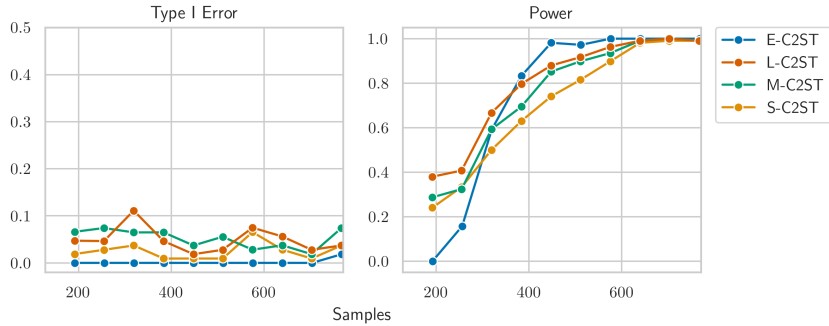

Figure 2: Power analysis and type I error for the KDEF data. All methods show very good power performance. The baselines start off with higher power. However, E-C2ST reaches power 1 the fastest while keeping the type I error lower than the baselines.

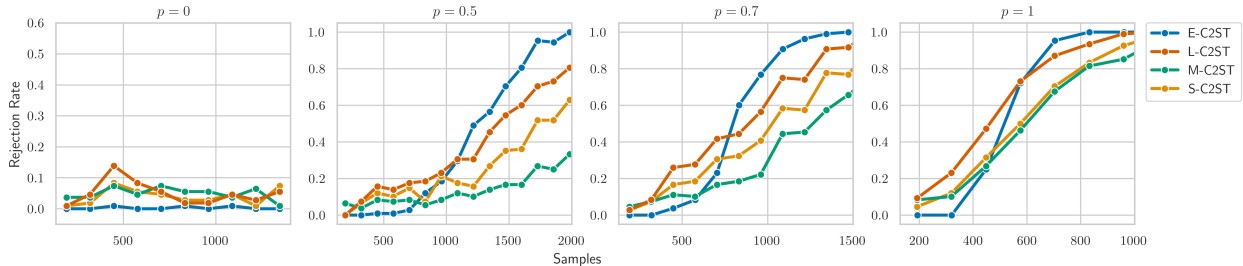

Figure 3: Power analysis for the Corrupted MNIST Data for different proportion of corruption $p = 0, 0.5, 0.7, 1$. Compared to the baselines, E-C2ST shows the highest power.

the alternative, we are given two sets of MNIST digits where one of them contains "corrupted" images, i.e. images generated from the trained DCGAN. We vary the portion of these images in the mini-batches to be $p = 0, 0.5, 0.7, 1$ and we resample the two datasets of size $3 \cdot 64, 4 \cdot 64, \ldots$. We fix the mini-batch size to be 64 for training E-C2ST and we conduct 100 independent runs. The rejection rates per sample size are displayed in Figure 3 for the four different cases. Note that the rejection rate in the case $p = 0$ refers to the estimated type I error. We can see that E-C2ST has superior power across the three levels of corruption compared to the baseline methods while keeping the type I error strictly below the significance level.

**The mini-batch size.** Here, we aim to determine the average number of samples required to effectively reject the null hypothesis. We perform power experiments on two datasets: MNIST ($p = 1$) and KDEF, using different batch sizes of 8, 16, 32, 64, and 128. Our methodology differs from previous experiments as we use a sequential testing approach. We continuously sample new batches and stop only when the null hypothesis is rejected. This procedure allows for dynamic adjustments of the sample size needed for maximum power.

We illustrate our results in Figure 4 using 100 independent experiments per scenario. The lines in Figure 4 represent the estimated power. Our results show that smaller batch sizes (e.g., 16 or 32) lead to faster rejection of the null hypothesis in terms of the number of samples, but not in terms of the number of batches required. Conversely, larger sample sizes require more samples to reject the null hypothesis, but reduce the number of steps involved, implying less computational power. For example, in the KDEF scenario, maximum power is achieved in only three steps when the batch size is 128. However, reducing batch size too strongly (e.g. batch size 8) can lead to reduced power (see MNIST case). We believe this is a result of training instabilities for very small batch sizes, leading to suboptimal networks at each step and thus small E-values.

**The initial value of $\lambda$.** We conducted power experiments on two datasets: MNIST (with $p = 1$) and KDEF, keeping the batch size constant at 32 samples. We varied $\lambda$ over 0.1, 0.3, 0.5, 0.7 and 0.9. We aim to determine the average number of samples required to effectively reject the null hypothesis. As in the previous experiment, we use a sequential testing strategy.

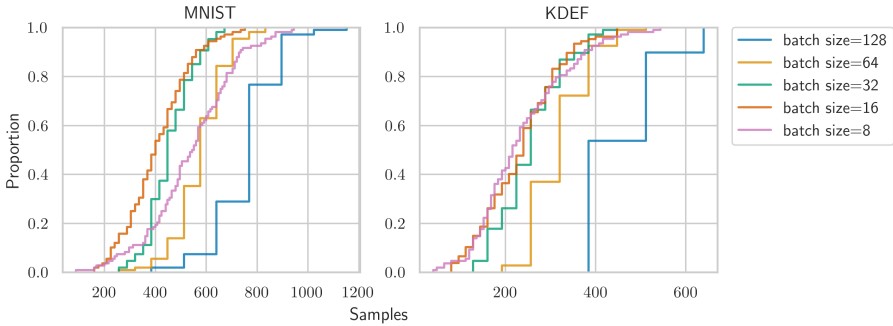

Figure 4: Power experiments performed on the MNIST and KDEF datasets using different batch sizes (8, 16, 32, 64, 128). The lines indicate the estimated power. In general, we can conclude that smaller batch sizes (with the exception of very small batches) allow faster rejection of the null hypothesis in terms of number of samples, and larger batch sizes require fewer steps but more samples.

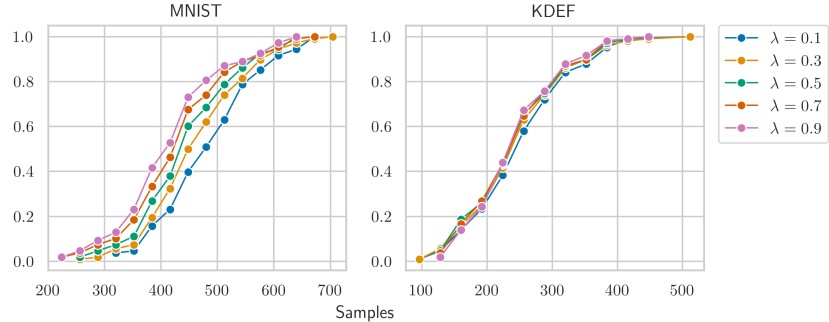

Figure 5: Power experiments performed on the MNIST and KDEF datasets for varying $\lambda = 0.1, 0.3, 0.5, 0.7, 0.9$ and fixed batch size of 32 samples. The lines indicate the estimated power. The initial value of $\lambda$ had no significant impact on the test performance in the KDEF scenario, while in the MNIST case, higher $\lambda$ values increased the test performance. This effect is due to the early stages of testing, where lower initial $\lambda$ values and the suboptimal neural network performance lead to lower batch E-values.

We visualized the results using 100 independent experiments for each scenario, as shown in Figure 5. The lines represent the power of the tests. From these results, we observe that the initial value of $\lambda$ does not significantly affect the power of the test in the KDEF scenario. However, in the MNIST case, we observe a more visible effect: higher values of $\lambda$ seem to increase the power of the test. This occurs because in the early stages of testing when sample sizes are smaller, the neural network tends to show suboptimal test performance. This is particularly evident when the initial $\lambda$ is low, resulting in lower conditional E-values. They affect the overall E-value, potentially resulting in reduced power performance. However, it is important to note that the initial $\lambda$ value does not drastically affect the number of samples required to achieve maximum power in our tests.

To summarize the two ablation studies, we compared the best E-C2ST performer based on the last two experiments with the baseline methods in Figure 7 in the Appendix, where we observe a significant gain in terms of power compared to the initial E-C2ST.

## 7 Discussion

We present E-CS2T, a deep E-value-based classifier two sample test tailored for both fixed and streaming data testing scenarios. This method combines predictive conditional independence tests with M-split likelihood ratio tests, resulting in a consistent test based on a E-variable that guarantees finite sample batch-wise anytime type I error control. Through empirical evaluations, we demonstrate that E-CS2T outperforms traditional $p$-value-based methods in terms of power, effectively utilizing the data and maintaining type I

error below the specified significance level. In addition, our observations highlight a trade-off between batch size and computational efficiency, and suggest an interesting future direction for exploring this trade-off from a theoretical and practical perspective.

**Online learning.** Even though our test is computationally efficient for large data sets, for small data regimes it is more expensive than the considered baseline methods (see Table 1). A promising direction for future work is to integrate an online training procedure into our method. By doing so, we could significantly reduce the computational time and make it linearly proportional to the number of batches processed.

**Active Learning.** Another interesting direction for future work is to use active learning techniques to improve the performance of our method. For example, we could prune each batch before including it in the training set by actively selecting the most informative samples. With this strategy, we could potentially improve the learning efficiency and effectiveness of the test.

## Acknowledgments

We would like to thank Peter Grünwald for his exciting talk at the AI4Science Colloquium, which introduced us to the theory and background of E-variables and safe testing. He inspired us to learn more about these topics and to pursue this research project.

Tim Bakker is partially supported by the Efficient Deep Learning research program, which is financed by the Dutch Research Council (NWO) in the domain "Applied and Engineering Sciences" (TTW).

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

## A  Proofs

**Proof of Lemma 2.2**

*Proof.* By Fubini's theorem we get:

$$\mathbb{E}_{\theta,\psi}\left[E^{(3)}\Big|z\right]$$

$$= \int E^{(3)}(x,y|z)\,(P_\theta \otimes P_\psi)\,(dx,dy|z)$$

$$= \int\int E^{(2)}(x|y,z) \cdot E^{(1)}(y|z)$$

$$P_\theta(dx|y,z)\,P_\psi(dy|z)$$

$$= \int\left(\int E^{(2)}(x|y,z)\,P_\theta(dx|y,z)\right)$$

$$\cdot\, E^{(1)}(y|z)\,P_\psi(dy|z)$$

$$\leq \int 1 \cdot E^{(1)}(y|z)\,P_\psi(dy|z) \leq 1. \qquad \square$$

**Lemma A.1** (Convex combinations of conditional E-variables). *If $\overline{E}$ and $\underline{E}$ are two conditional E-variables w.r.t. $\mathcal{H}_0 \subseteq \mathcal{P}(\mathcal{X})^{\mathcal{Z}}$ and $g : \mathcal{Z} \to [0,1]$ a measurable map, then:*

$$\tilde{E}(x|z) := g(z) \cdot \overline{E}(x|z) + (1 - g(z)) \cdot \underline{E}(x|z),$$

*also defines a conditional E-variable w.r.t. $\mathcal{H}_0$.*

**Proof of Lemma A.1**

$$\mathbb{E}_\theta\left[\tilde{E}\big|z\right]$$

$$= \int \tilde{E}(x|z)\,P_\theta(dx|z)$$

$$= \int \left(g(z) \cdot \overline{E}(x|z) + (1 - g(z)) \cdot \underline{E}(x|z)\right) P_\theta(dx|z)$$

$$= g(z)\int \overline{E}(x|z)P_\theta(dx|z) + (1 - g(z))\int \underline{E}(x|z)P_\theta(dx|z)$$

$$\leq g(z) \cdot \mathbf{1} + (1 - g(z)) \cdot \mathbf{1} = \mathbf{1}$$

**Proof of** (2) **being an e-variable**

*Proof.* For $\theta \in \Theta_0$ we have:

$$\mathbb{E}_\theta\left[E^{(m)}\Big|z^{(<m)}\right]$$

$$= \int E^{(m)}(z^{(m)}|z^{(<m)})\,p_\theta(y^{(m)}|y^{(<m)})\,\mu(dz^{(m)})$$

$$= \int \frac{p_{\mathsf{A}}(y^{(m)}|x^{(m)}, z^{(<m)})}{\max_{\tilde{\theta}\in\Theta_0} p_{\tilde{\theta}}(y^{(m)}|y^{(<m)})}\,p_\theta(y^{(m)}|y^{(<m)})\,\mu(dz^{(m)})$$

$$\leq \int \frac{p_{\mathsf{A}}(y^{(m)}|x^{(m)}, z^{(<m)})}{p_\theta(y^{(m)}|y^{(<m)})}\,p_\theta(y^{(m)}|y^{(<m)})\,\mu(dz^{(m)})$$

$$= \int p_{\mathsf{A}}(y^{(m)}|x^{(m)}, z^{(<m)})\,\mu(dz^{(m)}) = 1. \qquad \square$$

**Lemma A.1.** *Let $E_1, \ldots, E_D$ are $D$ E-variables with respect to the same null hypothesis class $\mathcal{H}_0$. Then, the average over all $D$ E-variables is an E-variable $\bar{E} := \frac{1}{D} \sum_{i=1}^{D} E_i$ is an E-variable.*

*Proof.* Let $p_\theta \in \mathcal{H}_0$. Then, for $\bar{E}$ we get

$$\mathbb{E}_\theta\left[\bar{E}\right] = \int \left(\frac{1}{D} \sum_{i=1}^{D} E_i(x)\right) p_\theta(x)\mu(dx) = \frac{1}{D} \sum_{i=1}^{D} \int E_i(x)p_\theta(x)\mu(dx) \leq \frac{1}{D} \sum_{i=1}^{D} 1 = 1$$

$\square$

**Proof of Corollary 3.2.** First, we here shortly review Ville's inequality:

**Theorem A.2** (Ville's Inequality, see Ville (1939)). *Let $(S^{(M)})_{M \in \mathbb{N}}$ be a non-negative supermartingale, $S^{(M)} : (\Omega, \mathcal{B}_\Omega, P) \to [0, \infty]$, $M \in \mathbb{N}$, w.r.t. filtration $\mathcal{F} = (\mathcal{F}_M)_{M \in \mathbb{N}}$, $\mathcal{F}_M \subseteq \mathcal{B}_\Omega$. Then for every $s > 1$ we have the inequality:*

$$P\left(\exists M \in \mathbb{N}. \, S^{(M)} \geq s\right) \leq \frac{\mathbb{E}[S^{(1)}]}{s}.$$

*Proof.* The sequence $\left(E^{(\leq M)}\right)_{M \in \mathbb{N}}$ constitutes a non-negative super-martingale of E-variables w.r.t. the filtration $\mathcal{F} := \left(\sigma(X^{(\leq M)})\right)$ due to the following computation for $P_\theta \in \mathcal{H}_0$:

$$\mathbb{E}_\theta\left[E^{(\leq M+1)} \,\middle|\, x^{(\leq M)}\right] = \int \prod_{m=1}^{M+1} E(X^{(m)}|x^{(<m)}) \, \mu(dx^{(M+1)})$$

$$= \int E(X^{(M+1)}|x^{(<M+1)}) \prod_{m=1}^{M} E(X^{(m)}|x^{(<m)}) \, \mu(dx^{(M+1)})$$

$$= \prod_{m=1}^{M} E(X^{(m)}|x^{(<m)}) \int E(X^{(M+1)}|x^{(<M+1)}) \, \mu(dx^{(M+1)})$$

$$\leq \prod_{m=1}^{M} E(X^{(m)}|x^{(<m)}) \cdot \mathbf{1}$$

$$= E^{(\leq M)}(x^{(\leq M)}) \cdot 1.$$

By Ville's inequality, see Thm. A.2, we get:

$$P_\theta\left(\exists M \in \mathbb{N}. \, E_\theta^{(\leq M)} \geq \alpha^{-1}\right) \leq \alpha.$$

$\square$

## B   Type II Error Control

A general finite sample bound for the type II error of testing based on the product of (conditional) i.i.d. E-variables can be achieved by Sanov's theorem, see Csiszár (1984); Balsubramani (2020).

**Theorem B.1** (Conditional type II error control for conditional i.i.d. E-variables). *Let $E : \mathcal{X} \times \mathcal{Z} \to \mathbb{R}_{\geq 0}$ be a conditional E-variable w.r.t. $\mathcal{H}_0$ given $\mathcal{Z}$. Let $X_1, \ldots, X_N : \Omega \times \mathcal{Z} \to \mathcal{X}$ be conditional random variables that are i.i.d. conditioned on $\mathcal{Z}$. Let $E^{(N)} := \prod_{n=1}^{N} E(X_n|Z)$. Let $\alpha \in (0,1]$, $\gamma_N := -\frac{1}{N}\log\alpha \geq 0$ and for $\gamma \in \mathbb{R}_{\geq 0}$ put:*

$$\mathcal{A}_\gamma^{|z} := \{Q \in \mathcal{P}(\mathcal{X}) \,|\, \mathbb{E}_{X \sim Q}[\log E(X|z)] \leq \gamma\}.$$

*Then for every $P_\theta \in \mathcal{H}_A$ and $z \in \mathcal{Z}$ we have the following type II error bound:*

$$P_\theta\left(E^{(N)} \leq \alpha^{-1}\Big|Z = z\right) \leq \exp\left(-N \cdot \mathrm{KL}(\mathcal{A}_{\gamma_N}^{|z}\|P_\theta^{|z})\right),$$

*which converges to 0 if $\mathrm{KL}(\mathcal{A}_\gamma^{|z}\|P_\theta^{|z}) > 0$ for some $\gamma > 0$. Note that for a subset $\mathcal{A} \subseteq \mathcal{P}(\mathcal{X})$ we abbreviate:*

$$\mathrm{KL}(\mathcal{A}\|P) := \inf_{Q \in \mathcal{A}} \mathrm{KL}(Q\|P).$$

*Proof.* If $\hat{P}_N := \frac{1}{N}\sum_{n=1}^{N} \delta_{X_n|Z}$ is the empirical distribution then we get the following equivalence, when conditioned on $Z = z$:

$$E^{(N)|z} \leq \alpha^{-1} \iff \prod_{n=1}^{N} E(X_n|z) \leq \alpha^{-1}$$

$$\iff \frac{1}{N}\sum_{n=1}^{N}\log E(X_n|z) \leq -\frac{1}{N}\log\alpha =: \gamma_N$$

$$\iff \mathbb{E}_{X \sim \hat{P}_N^{|z}}[\log E(X|z)] \leq \gamma_N$$

$$\iff \hat{P}_N^{|z} \in \mathcal{A}_{\gamma_N}^{|z}.$$

The bound then follows by a simple application of Sanov's theorem, see Csiszár (1984); Balsubramani (2020), for each $z \in \mathcal{Z}$ individually:

$$P_\theta\left(E^{(N)} \leq \alpha^{-1}\Big|Z = z\right) = P_\theta\left(\hat{P}_N \in \mathcal{A}_{\gamma_N}^{|z}\Big|Z = z\right) \leq \exp\left(-N \cdot \mathrm{KL}(\mathcal{A}_{\gamma_N}^{|z}\|P_\theta^{|z})\right),$$

which requires the i.i.d. assumption (conditioned on $Z$) and that $\mathcal{A}_{\gamma_N}^{|z}$ is completely convex, which it is.   $\square$

**Lemma B.2.** *Consider the situation in Theorem B.1 and fix $z \in \mathcal{Z}$. Then the first statement implies the second:*

1. $\mathrm{KL}(\mathcal{A}_{\gamma(z)}^{|z}\|P_\theta^{|z}) > 0$ *for some $\gamma(z) > 0$.*

2. $\mathbb{E}_{X \sim P_\theta^{|z}}[\log E(X|z)] > 0$.

*If, furthermore, $\sup_{x \in \mathcal{X}} |\log E(x|z)| < \infty$ then the set $\mathcal{A}_{\gamma(z)}^{|z}$ is TV-closed in $\mathcal{P}(\mathcal{X})$ for every $\gamma(z) \geq 0$. In this case, the second statement also implies the first one, where we then have the implication:*

$$0 \leq \gamma(z) < \mathbb{E}_{X \sim P_\theta^{|z}}[\log E(X|z)] \quad \implies \quad \mathrm{KL}(\mathcal{A}_{\gamma(z)}^{|z}\|P_\theta^{|z}) > 0.$$

*Proof.* "$\implies$": If $\mathbb{E}_{X \sim P_\theta^{|z}}[\log E(X|z)] \leq 0$ then $P_\theta^{|z} \in \mathcal{A}_0$. Since $\mathcal{A}_0 \subseteq \mathcal{A}_\gamma$ for every $\gamma > 0$ we get: $\mathrm{KL}(\mathcal{A}_\gamma^{|z}\|P_\theta^{|z}) = 0$ for every $\gamma > 0$.

"$\Longleftarrow$": Assume $C := \sup_{x \in \mathcal{X}} |\log E(x|z)| < \infty$ and $\gamma \geq 0$. Let $Q \in \mathcal{P}(\mathcal{X})$ be a TV-limit point of a sequence $P_n \in \mathcal{A}_\gamma^{|z}$, $n \in \mathbb{N}$. Then we have the inequality:

$$\begin{aligned}
\mathbb{E}_{X \sim Q}[\log E(X|z)] &= \mathbb{E}_{X \sim Q}[\log E(X|z)] - \mathbb{E}_{X \sim P_n}[\log E(X|z)] + \mathbb{E}_{X \sim P_n}[\log E(X|z)] \\
&\leq |\mathbb{E}_{X \sim Q}[\log E(X|z)] - \mathbb{E}_{X \sim P_n}[\log E(X|z)]| + \mathbb{E}_{X \sim P_n}[\log E(X|z)] \\
&\leq C \cdot \mathrm{TV}(Q, P_n) + \gamma \\
&\to \gamma.
\end{aligned}$$

This shows that $Q \in \mathcal{A}_\gamma^{|z}$ as well, and, thus, $\mathcal{A}_\gamma^{|z}$ is TV-closed. By way of contradiction now assume that $\mathbb{E}_{X \sim P_\theta^{|z}}[\log E(X|z)] > 0$, but $\mathrm{KL}(\mathcal{A}_\gamma^{|z} \| P_\theta^{|z}) = 0$ for all $\gamma > 0$. Since $\mathcal{A}_\gamma^{|z}$ is TV-closed and (completely) convex we have that $P_\theta^{|z} \in \mathcal{A}_\gamma^{|z}$ for all $\gamma > 0$. So we get:

$$\mathbb{E}_{X \sim P_\theta^{|z}}[\log E(X|z)] \leq \gamma,$$

for all $\gamma > 0$, and thus: $\mathbb{E}_{X \sim P_\theta^{|z}}[\log E(X|z)] \leq 0$, which is a contradiction to our assumption. $\qquad\square$

The unconditional version follows from the above by using the one-point space $\mathcal{Z} = \{*\}$ and reads like:

**Corollary B.3** (Type II error control for i.i.d. E-variables)**.** *Let $X_1, \ldots, X_N$ be an i.i.d. sample, $E : \mathcal{X} \to \mathbb{R}_{\geq 0}$ be an E-variable w.r.t. $\mathcal{H}_0$ and $E^{(N)} := \prod_{n=1}^N E(X_n)$. Let $\alpha \in (0, 1]$, $\gamma_N := -\frac{1}{N} \log \alpha \geq 0$ and for $\gamma \in \mathbb{R}_{\geq 0}$ put:*

$$\mathcal{A}_\gamma := \{Q \in \mathcal{P}(\mathcal{X}) \,|\, \mathbb{E}_Q[\log E] \leq \gamma\}.$$

*Then for every $P_\theta \in \mathcal{H}_A$ we have the following type II error bound:*

$$P_\theta \left( E^{(N)} \leq \alpha^{-1} \right) \leq \exp\left(-N \cdot \mathrm{KL}(\mathcal{A}_{\gamma_N} \| P_\theta)\right),$$

*which converges to 0 if $\mathrm{KL}(\mathcal{A}_\gamma \| P_\theta) > 0$ for some $\gamma > 0$.*

Relating to the simpler unconditional case of the Corollary we can make the following clarifying remarks.

**Remark B.4.** *1. The condition: $\mathrm{KL}(\mathcal{A}_\gamma \| P_\theta) > 0$ for some $\gamma > 0$, is slightly stronger than the condition: $\mathbb{E}_{P_\theta}[\log E] > 0$. If $\sup_{x \in \mathcal{X}} |\log E(x)| < \infty$ then one can show that both those conditions are equivalent.*

*2. If there exist $\delta, \gamma > 0$ such that for all $P_\theta \in \mathcal{H}_A$ we have $\mathrm{KL}(\mathcal{A}_\gamma \| P_\theta) \geq \delta$ then we easily deduce the uniform type II error bound for $N \geq -\frac{\log \alpha}{\gamma}$:*

$$\sup_{P_\theta \in \mathcal{H}_A} P_\theta \left( E^{(N)} \leq \alpha^{-1} \right) \leq \exp\left(-N \cdot \delta\right).$$

From Theorem B.1 we can also get a type II error control for conditional i.i.d. E-variables if we assume that the distribution for the conditioning variable is a marginal part of the hypothesis.

**Corollary B.5** (Unconditional type II error control for conditional i.i.d. E-variables)**.** *Let $E : \mathcal{X} \times \mathcal{Z} \to \mathbb{R}_{\geq 0}$ be a conditional E-variable w.r.t. $\mathcal{H}_0$ given $\mathcal{Z}$. Let $Z : \Omega \to \mathcal{Z}$ be a fixed random variable with values in $\mathcal{Z}$ and let $X_1, \ldots, X_N : \Omega \to \mathcal{X}$ be random variables that are i.i.d. conditioned on $Z$. Let $E^{(N)} := \prod_{n=1}^N E(X_n|Z)$. Let $\alpha \in (0, 1]$, $\gamma_N := -\frac{1}{N} \log \alpha \geq 0$. Then for every $P_\theta \in \mathcal{H}_A$ we have the following type II error bound:*

$$P_\theta \left( E^{(N)} \leq \alpha^{-1} \right) \leq \mathbb{E}_\theta \left[ \exp\left(-N \cdot \mathrm{KL}(\mathcal{A}_{\gamma_N}^{|Z} \| P_\theta^{|Z})\right) \right]$$

$$\leq \exp\left(-N \cdot \inf_{z \in \mathcal{Z}} \mathrm{KL}(\mathcal{A}_{\gamma_N}^{|z} \| P_\theta^{|z})\right),$$

*where the middle term converges to 0 for $N \to \infty$ if for $P_\theta(Z)$-almost-all $z \in \mathcal{Z}$ there exists some $\gamma > 0$ such that $\mathrm{KL}(\mathcal{A}_\gamma^{|z} \| P_\theta^{|z}) > 0$. The latter is e.g. the case if $\inf_{z \in \mathcal{Z}} \mathrm{KL}(\mathcal{A}_\gamma^{|z} \| P_\theta^{|z}) > 0$ for some $\gamma > 0$.*

*Proof.* The inequalities directly follow from Theorem B.1 by plugging the random variable $Z$ into $z$ and taking expectation values:

$$
\begin{aligned}
P_\theta\left(E^{(N)} \le \alpha^{-1}\right) &= \mathbb{E}_\theta\left[P_\theta\left(E^{(N)} \le \alpha^{-1}\Big|Z\right)\right] \\
&\le \mathbb{E}_\theta\left[\exp\left(-N \cdot \mathrm{KL}(\mathcal{A}_{\gamma_N}^{|Z}\|P_\theta^{|Z})\right)\right] \\
&\le \sup_{z \sim P_\theta(Z)} \exp\left(-N \cdot \mathrm{KL}(\mathcal{A}_{\gamma_N}^{|z}\|P_\theta^{|z})\right) \\
&= \exp\left(-N \cdot \inf_{z \sim P_\theta(Z)} \mathrm{KL}(\mathcal{A}_{\gamma_N}^{|z}\|P_\theta^{|z})\right) \\
&\le \exp\left(-N \cdot \inf_{z \in \mathcal{Z}} \mathrm{KL}(\mathcal{A}_{\gamma_N}^{|z}\|P_\theta^{|z})\right).
\end{aligned}
$$

Here $\sup_{z \sim P_\theta(Z)}$ and $\inf_{z \sim P_\theta(Z)}$ denote the essential supremum, essential infimum, resp., w.r.t. $P_\theta(Z)$.

The statement of the convergence follows from the dominated convergence theorem and the observation that for every $z \in \mathcal{Z}$ we have the trivial bounds:

$$
0 \le \exp\left(-N \cdot \mathrm{KL}(\mathcal{A}_{\gamma_N}^{|z}\|P_\theta^{|z})\right) \le 1.
$$

This shows the claim. □

### B.1 Type II Error M=2

**Theorem B.6** (Type-II error control for conditional E-variable for singleton $\mathcal{H}_0$)**.** *Let $\mathcal{H}_0 = \{P_0\}$ be a singleton set. Consider a model class $\mathcal{H}_\mathsf{A}$ and a learning algorithm that for every realization $x = (x_n)_{n \in \mathbb{N}} \in \mathcal{X}^{\mathbb{N}}$ and every number $N^{(1)} \in \mathbb{N}$ fits a model $P_\mathsf{A}^{|x^{(1)}} \in \mathcal{P}(\mathcal{X})$ to the first $N^{(1)}$ entries $x^{(1)} = (x_n)_{n \in \mathcal{I}^{(1)}}$ of $x$. Assume that for every $P_\theta \in \mathcal{H}_\mathsf{A}$ and $P_\theta$-almost every i.i.d. realization $x = (x_n)_{n \in \mathbb{N}}$ of $P_\theta$ there exists a number $N^{(1)}(x) \in \mathbb{N}$ and $\epsilon(x) > 0$ such that for all $N^{(1)} \ge N^{(1)}(x)$ the model $P_\mathsf{A}^{|x^{(1)}} \in \mathcal{P}(\mathcal{X})$ has a density $p_\mathsf{A}(x_n|x^{(1)})$ and satisfies:*

$$
\mathrm{KL}(P_\theta\|P_\mathsf{A}^{|x^{(1)}}) < \mathrm{KL}(P_\theta\|P_0) - \epsilon(x), \qquad\qquad \sup_{x_n \in \mathcal{X}} |\log E(x_n|x^{(1)})| < \infty.
$$

*Then for every $N^{(1)}, N^{(2)} \in \mathbb{N}$ we have the bound:*

$$
P_\theta\left(E^{(N^{(2)}|N^{(1)})} \le \alpha^{-1}\right) \le \mathbb{E}_{X^{(1)} \sim P_\theta}\left[\exp\left(-N^{(1)} \cdot \mathrm{KL}(\mathcal{A}_{\gamma_{N^{(2)}}}^{|X^{(1)}}\|P_\theta)\right)\right],
$$

*which converges to zero for $\min(N^{(1)}, N^{(2)}) \to \infty$.*

*Proof.* The bound directly follows from Corollary B.5. Note that by the independence assumptions, we have $P_\theta^{|x^{(1)}} = P_\theta$. Then note that for $P_\theta$-almost-all $x \in \mathcal{X}^{\mathbb{N}}$ and for $N^{(1)} \ge N^{(1)}(x)$:

$$
\mathbb{E}_{X_n \sim P_\theta}\left[\log E(X_n|x^{(1)})\right] = \mathrm{KL}(P_\theta\|P_0) - \mathrm{KL}(P_\theta\|P_\mathsf{A}^{|x^{(1)}}) > \epsilon(x) > 0.
$$

By assumption and Lemma B.2 we have that for $P_\theta$-almost all $x \in \mathcal{X}^{\mathbb{N}}$ and for $N^{(1)} > N^{(1)}(x)$ and for $\gamma(x^{(1)})$ with:

$$
0 < \gamma(x^{(1)}) < \epsilon(x) < \mathbb{E}_{X_n \sim P_\theta}\left[\log E(X_n|x^{(1)})\right]
$$

we have: $\mathrm{KL}(\mathcal{A}_{\gamma(x^{(1)})}^{|x^{(1)}}\|P_\theta) > 0$. So for $N^{(2)}$ big enough we get:

$$
\gamma_{N^{(2)}} := -\frac{1}{N^{(2)}} \log \alpha < \epsilon(x).
$$

This shows that for:

$$N^{(2)} > \frac{-\log \alpha}{\epsilon(x)} =: N^{(2)}(x),$$

we have: $\mathrm{KL}(\mathcal{A}^{|x^{(1)}}_{\gamma_{N^{(2)}}} \| P_\theta) > 0$. This shows that for $P_\theta$-almost-all $x \in \mathcal{X}^{\mathbb{N}}$ we have that:

$$\exp\left(-N^{(2)} \cdot \mathrm{KL}(\mathcal{A}^{|x^{(1)}}_{\gamma_{N^{(2)}}} \| P_\theta)\right) \longrightarrow 0, \qquad \text{for} \qquad \min(N^{(1)}, N^{(2)}) \to \infty.$$

Since we always have the trivial bounds:

$$0 \le \exp\left(-N^{(2)} \cdot \mathrm{KL}(\mathcal{A}^{|x^{(1)}}_{\gamma_{N^{(2)}}} \| P_\theta)\right) \le 1,$$

the theorem of dominated convergence tells us that we also have the convergence:

$$\mathbb{E}_{X^{(1)} \sim P_\theta}\left[\exp\left(-N^{(2)} \cdot \mathrm{KL}(\mathcal{A}^{|X^{(1)}}_{\gamma_{N^{(2)}}} \| P_\theta)\right)\right] \longrightarrow 0, \qquad \text{for} \qquad \min(N^{(1)}, N^{(2)}) \to \infty.$$

This shows the claim. $\qquad \square$

**Lemma B.7.** *Let* $\tilde{P}_{\mathsf{A}}(y|x, x^{(1)}, y^{(1)}) = (1 - \lambda) P_{\mathsf{A}}(y|x, x^{(1)}, y^{(1)}) + \lambda \cdot P_0(y|y^{(2)})$ *for* $\lambda \in (0,1)$ *and* $y \in \{0,1\}$. *Then the conditional E-variable defined by*

$$\tilde{E}(x, y|x^{(1)}, y^{(1)}) = \frac{\tilde{p}_{\mathsf{A}}(y|x, x^{(1)}, y^{(1)})}{p_0(y|y^{(2)})} = \lambda + (1 - \lambda)E(x, y|x^{(1)}, y^{(1)}) \tag{11}$$

*is bounded, i.e.* $\|\log \tilde{E}\|_\infty < \infty$.

*Proof.* For every $\mathcal{I}^{(2)} \subset \mathcal{X} \times \mathcal{Y}$ and every $(x, y) \in \mathcal{I}^{(2)}$

$$\log \tilde{E}(x, y|x^{(1)}, y^{(1)}) = \log\left(\lambda + (1 - \lambda)E(x, y|x^{(1)}, y^{(1)})\right) \ge \log \lambda$$

and

$$\log \tilde{E}(x, y|x^{(1)}, y^{(1)}) = \log\left(\lambda + (1 - \lambda)E(x, y|x^{(1)}, y^{(1)})\right)$$
$$\le \log\left(\lambda + \frac{1 - \lambda}{\min_{\mathcal{I}^{(2)} \subset \mathcal{X} \times \mathcal{Y}} p_0(y|y^{(2)})}\right)$$
$$\le \log\left(\lambda + \frac{1 - \lambda}{1/N^{(2)}}\right) = \log(\lambda + (1 - \lambda)N^{(2)})$$

$\qquad \square$

**Corollary B.8.** *Let the assumptions about the learner in Theorem B.6 hold, i.e. for every realization* $x = (x_n)_{n \in \mathbb{N}} \in \mathcal{X}^{\mathbb{N}}$ *and every number* $N^{(1)} \in \mathbb{N}$ *the learner fits a model* $P_{\mathsf{A}}^{|x^{(1)}} \in \mathcal{P}(\mathcal{X})$ *to the first* $N^{(1)}$ *entries* $x^{(1)} = (x_n)_{n \in \mathcal{I}^{(1)}}$ *of* $x$. *Assume that for every* $P_\theta \in \mathcal{H}_{\mathsf{A}}$ *and* $P_\theta$*-almost every i.i.d. realization* $x = (x_n)_{n \in \mathbb{N}}$ *of* $P_\theta$ *there exists a number* $N^{(1)}(x) \in \mathbb{N}$ *and* $\epsilon(x) > 0$ *such that for all* $N^{(1)} \ge N^{(1)}(x)$ *the model* $P_{\mathsf{A}}^{|x^{(1)}} \in \mathcal{P}(\mathcal{X})$ *has a density* $p_{\mathsf{A}}(x_n|x^{(1)})$ *and satisfies:*

$$\mathrm{KL}(P_\theta \| P_{\mathsf{A}}^{|x^{(1)}}) < \mathrm{KL}(P_\theta \| P_0) - \epsilon(x),$$

*Then the statististical test w.r.t.* $\tilde{E}(x, y|x^{(1)}, y^{(1)})$ *from Equation 11 is consistent.*

*Proof.* The claim follows directly from Theorem B.6 since $\tilde{E}(x, y|x^{(1)}, y^{(1)})$ is bounded according to Lemma B.7.
$\qquad \square$

## B.2 Type II Error Control $M = \infty$

**Theorem B.9** (Strong Law of Large Numbers for Martingale Difference Sequences). *Consider the probability space $(\Omega, \mathcal{F}, P)$. Let $(X_n)_{n \in \mathbb{N}}$ be a sequence random variables that satisfies for some $r \geq 1$*

$$\sum_{n=1}^{\infty} \frac{\mathbb{E}_P[|X_n|^{2r}]}{n^{r+1}} < 0$$

*Consider the natural filtration $\mathcal{F}_n = \sigma(X_1, \ldots, X_n) \subset \mathcal{F}$. Additionally, let $\mathbb{E}_P[X_n | \mathcal{F}_{n-1}] = 0$ for all $n \in \mathbb{N}$. Then, it holds $\frac{1}{n} \sum_{k=1}^{n} X_k \overset{a.s.}{\to} 0$.*

**Proof of Theorem 3.4**

*Proof.* Consider the stopping time $T$ adapted to the natural filtration $\mathcal{F}_M = \sigma(X^{(\leq M)}\})$ given by

$$T = \inf\{M \geq 0 : E^{(\leq M)} \geq 1/\alpha\}.$$

For this stopping time we have the equivalent relation

$$T < \infty \iff \exists M \geq 0 \text{ such that } E^{(\leq M)} \geq \alpha^{-1}$$

The probability that the null distribution will not be rejected in favor of the alternative is given by

$$P_\theta(T = \infty) = P_\theta(\forall M \geq 0 : E^{(\leq M)} < \alpha^{-1}) = P_\theta\left(\forall M \geq 0 : \frac{1}{M} \sum_{m=1}^{M} \log E^{(m)} < \frac{\log \alpha^{-1}}{M}\right)$$

Next, define the random variables $W_m$ :

$$W_m = \mathbb{E}_\theta[\log E^{(m)} | \mathcal{F}_{m-1}] = \mathbb{E}_\theta\left[\log\left(\prod_{i \in \mathcal{I}^{(m)}} E(x_i | x^{(<m)})\right) \Big| \mathcal{F}_{m-1}\right]$$

$$= \mathbb{E}_\theta\left[\log \frac{p_\theta(x^{(m)})}{p_0(x^{(m)} | \hat{\theta}_0(x^{(m)}))}\right] - \mathrm{KL}(P_\theta \| P_\mathsf{A}^{|x^{(<m)}}) > r_m.$$

Then the above probability equals

$$P_\theta\left(\forall M \geq 0 : \frac{1}{M} \sum_{m=1}^{M} \log E^{(m)} < \frac{\log \alpha^{-1}}{M}\right) = P_\theta\left(\forall M \geq 0 : \frac{1}{M} \sum_{m=1}^{M} \log E^{(m)} - W_m + \frac{1}{M} \sum_{m=1}^{M} W_m < \frac{\log \alpha^{-1}}{M}\right)$$

$$\leq P_\theta\left(\forall M \geq 0 : \frac{1}{M} \sum_{m=1}^{M} \log E^{(m)} - W_m + \frac{1}{M} \sum_{m=1}^{M} r_m - \frac{\log \alpha^{-1}}{M} < 0\right)$$

Next, we will prove that $\frac{1}{M} \sum_{m=1}^{M} \log E^{(m)} - W_m \overset{a.s.}{\to} 0$. Note that $\sum_{m=1}^{M} \log E^{(m)} - W_m$ is a martingale w.r.t. the filtration $\mathcal{F}_M$ with bounded martingale differences $\log E^{(m)} - W_m$ in $L_2$. This results from the boundedness of $\log E^{(m)}$:

$$\mathbb{E}[(\log E^{(m)} - W_m)^2] \leq \sup_{(x_n)_{n \in \mathcal{I}(\leq m)}} (\log E^{(m)} - W_m)^2 \leq 4s_m^2$$

It follows that

$$\sum_{m=1}^{\infty} \frac{\mathbb{E}[(\log E^{(m)} - W_m)^2]}{m^2} \leq \sum_{m=1}^{\infty} \frac{4s_m^2}{m^2} < \infty$$

With Theorem B.9 we get that $\frac{1}{M}\sum_{m=1}^{M}\log E^{(m)} - W_m \overset{a.s.}{\to} 0$. This implies

$$\limsup_{M\to\infty} \frac{1}{M}\sum_{m=1}^{M}\log E^{(m)} - W_m - \frac{\log\alpha^{-1}}{M} + \frac{1}{M}\sum_{m=1}^{M}r_m = 0 - 0 + r > 0,$$

where $r = \limsup_{M\to\infty}\sum_{m=1}^{M}\frac{r_m}{M} > 0$. Note that the sequence can even diverge, i.e. $r = +\infty$. Thus,

$$P_\theta\left(\forall M \geq 0: \ \frac{1}{M}\sum_{m=1}^{M}\log E^{(m)} - W_m + \frac{1}{M}\sum_{m=1}^{M}r_m - \frac{\log\alpha^{-1}}{M} < 0\right)$$
$$\leq P_\theta\left(\limsup_{M\to\infty}\frac{1}{M}\sum_{m=1}^{M}\log E^{(m)} - W_m - \frac{\log\alpha^{-1}}{M} + \frac{1}{M}\sum_{m=1}^{M}r_m \leq 0\right) = 0.$$

It follows that $P_\theta(T = \infty) = 0$. □

**Proof of Lemma 5.1**

*Proof.* By Lemma B.7 it follows that the defined E-variable is bounded with bound depending on the batch size. With $|\mathcal{I}^{(m)}| < B$ for all $m$ the statement follows directly from Theorem 3.4 □

# C Expected Log Growth Rate

**Theorem C.1.** *Consider the sequence of E-C2ST E-variables $(E^{(\leq M)})_{M \geq 1}$ with increments $E^{(m)}$ for $m = 1, \ldots, M$ defined as in Equation (9). Let $P(X, Y)$ denote the true joint distribution of $X$ and $Y$ with probability density function $p(x, y) = p(y|x)p(x)$.*

*Then it holds for all $M \geq 1$*

$$\mathbb{E}_{X^{(\leq M)}, Y^{(\leq M)}} \left[ \log E^{(\leq M)} \right] \leq |\mathcal{I}^{(\leq M)}| \cdot I(X; Y)$$

*Proof.* For any $M \geq 1$ we get due to the independence of the observations:

$$\mathbb{E}_{X^{(\leq M)}, Y^{(\leq M)}} \left[ \log E^{(\leq M)} \right] = \sum_{m=1}^{M} \mathbb{E}_{X^{(\leq M)}, Y^{(\leq M)}} \left[ \log E^{(m)} \right] = \sum_{m=1}^{M} \mathbb{E}_{X^{(\leq m)}, Y^{(\leq m)}} \left[ \log E^{(m)} \right]$$

$$= \sum_{m=1}^{M} \mathbb{E}_{X^{(<m)}, Y^{(<m)}} \left[ \mathbb{E}_{X^{(m)}, Y^{(m)}} \left[ \log E^{(m)} | X^{(<m)}, Y^{(<m)} \right] \right]$$

Let $\tilde{P}_A(X, Y | X^{(<M)}, Y^{(<M)})$ be the estimated joint distribution of $X$ and $Y$ under the alternative with probability density function $\tilde{p}_A(x, y) = \tilde{p}_A(y|x, x^{(<M)}, y^{(<M)})p(x)$, where $p(x)$ is the unknown marginal distribution of $X$ and $p_A(y|x, x^{(<M)}, y^{(<M)})$ is the learner trained on $x^{(<M)}, y^{(<M)}$. We get for the increments

$$\mathbb{E}_{X^{(m)}, Y^{(m)}} \left[ \log E^{(m)} | X^{(<m)}, Y^{(<m)} \right] = \mathbb{E}_{X^{(m)}, Y^{(m)}} \left[ \log \frac{\tilde{p}_A(y^{(m)}|x^{(m)}, x^{(<m)}, y^{(<m)})}{p(y^{(m)}|\hat{\theta}_0(y^{(m)}))} \right]$$

$$= \mathbb{E}_{X^{(m)}, Y^{(m)}} \left[ \log \frac{\tilde{p}_A(y^{(m)}|x^{(m)}, x^{(<m)}, y^{(<m)})p(y^{(m)}|x^{(m)})p(y^{(m)})}{p(y^{(m)}|\hat{\theta}_0(y^{(m)}))p(y^{(m)}|x^{(m)})p(y^{(m)})} \right]$$

$$= \mathbb{E}_{X^{(m)}, Y^{(m)}} \left[ \log \frac{\tilde{p}_A(y^{(m)}|x^{(m)}, x^{(<m)}, y^{(<m)})}{p(y^{(m)}|x^{(m)})} \right] + \mathbb{E}_{X^{(m)}, Y^{(m)}} \left[ \log \frac{p(y^{(m)}|x^{(m)})}{p(y^{(m)})} \right] + \mathbb{E}_{X^{(m)}, Y^{(m)}} \left[ \log \frac{p(y^{(m)})}{p(y^{(m)}|\hat{\theta}_0(y^{(m)}))} \right]$$

$$= - \text{KL}(P^{(m)} \| \tilde{P}_A^{(m)|(<m)}) + |\mathcal{I}^{(m)}| \cdot I(X; Y) + \mathbb{E}_{Y^{(m)}} \left[ \log \frac{p(y^{(m)})}{p(y^{(m)}|\hat{\theta}_0(y^{(m)}))} \right]$$

where

$$\text{KL}(P^{(m)} \| \tilde{P}_A^{(m)|(<m)}) = - \mathbb{E}_{X^{(m)}, Y^{(m)}} \left[ \log \frac{\tilde{p}_A(y^{(m)}|x^{(m)}, x^{(<m)}, y^{(<m)})p(x^{(m)})}{p(y^{(m)}|x^{(m)})p(x^{(m)})} \right].$$

Plugging this into the above expression for $\mathbb{E}_{X^{(\leq M)}, Y^{(\leq M)}} \left[ \log E^{(\leq M)} \right]$ we get

$$\mathbb{E}_{X^{(\leq M)}, Y^{(\leq M)}} \left[ \log E^{(\leq M)} \right] = - \sum_{m} \mathbb{E}_{X^{(<m)}, Y^{(<m)}} \left[ \text{KL}(P^{(m)} \| \tilde{P}_A^{(m)|(<m)}) \right] + |\mathcal{I}^{(\leq M)}| \cdot I(X; Y)$$

$$+ \sum_{m=1}^{M} \mathbb{E}_{Y^{(m)}} \left[ \log \frac{p(y^{(m)})}{p(y^{(m)}|\hat{\theta}_0(y^{(m)}))} \right]$$

First, note that $p(y^{(m)}|\hat{\theta}_0(y^{(m)})) \geq p(y^{(m)})$ for any $m$ and thus $\mathbb{E}_{Y^{(m)}} \left[ \log \frac{p(y^{(m)})}{p(y^{(m)}|\hat{\theta}_0(y^{(m)}))} \right] \leq 0$. Together with the non-negativity of the KL we get that

$$\mathbb{E}_{X^{(\leq M)}, Y^{(\leq M)}} \left[ \log E^{(\leq M)} \right] \leq -0 + |\mathcal{I}^{(\leq M)}| \cdot I(X; Y) + 0 = |\mathcal{I}^{(\leq M)}| \cdot I(X; Y) (= Mb \cdot I(X; Y) \text{ for constant batch size})$$

$\square$

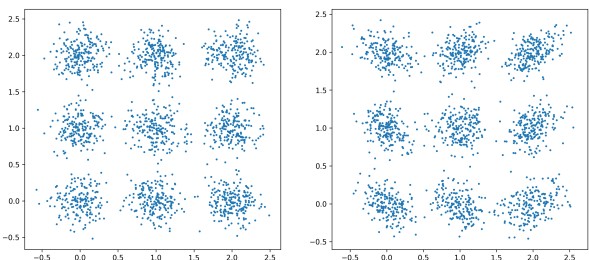

Figure 6: The two classes of the Blob dataset.

## D   Experiments

In this section, we explain the implementation and training of our models in detail. In Section D.2, we discuss the architecture choice and training of E-C2ST and the other baseline methods for the synthetic and image data experiments. Code will be provided upon acceptance.

### D.1   Baselines

We compare E-C2ST to the following baselines.

- **S-C2ST** (standard C2ST), is the C2ST proposed by Lopez-Paz and Oquab (2017). We train a binary classifier on the augmented data. The null hypothesis is that accuracy is 0.5 and the alternative is that it is larger 0.5. The $p$-values is computed via a permutation test.

- **L-C2ST** (logits C2ST) proposed by Cheng and Cloninger (2022) is a kernel based test, which again trains a binary classifier to distinguish the two classes. The null hypothesis is rejected if the difference between the classes logits average is not significant. The $p$-values is computed via a permutation test.

- **M-C2ST** We conduct the tests by means of the proposed test statistics based on maximum mean discrepancy (Kirchler et al., 2020). The $p$-values is computed via a permutation test.

### D.2   Training

We used Adam optimizer (Kingma and Ba, 2015) with learning rate $1 \cdot 1e - 4$ (and $5 \cdot 1e - 4$ for the Blob data). For fitting the parameter $\lambda$ from (10) we used L-BFGS-B (Byrd et al., 1995) implemented in (Virtanen et al., 2020) and we set the initial value to 0.5 unless specified otherwise. Note that in all experiments we consider a paired two-sample test for simplicity, i.e. each observation consists of a pair $X$ and $Y$ that possibly come from different observations.

- **Blob data.** The two Blob distributions used in the corresponding type 2 error experiment are visualized in Figure 6. The means are the same for both classes and are arranged in a $3 \times 3$ grid. The two populations differ in their variance. The used network architectures are described in Table 2. We trained the models with early stopping with patience 20 for all methods in all cases.

- **MNIST.** The dataset is obtained from `https://github.com/fengliu90/DK-for-TST`. Table 3 outlines the neural network architectures. We trained the models with early stopping with patience of 15 epochs for the baseline methods and 10 for E-C2ST.

- **Face Expression Data.** For all methods we used the network architecture provided in Table 4. We set the patience parameter to 20 epochs.

### D.3   Additional Experiments

**Best models according to the ablation study.**   In the main paper, we performed two experiments to investigate the effect of batch size and the effect of initial lambda value. Here we summarize our results by

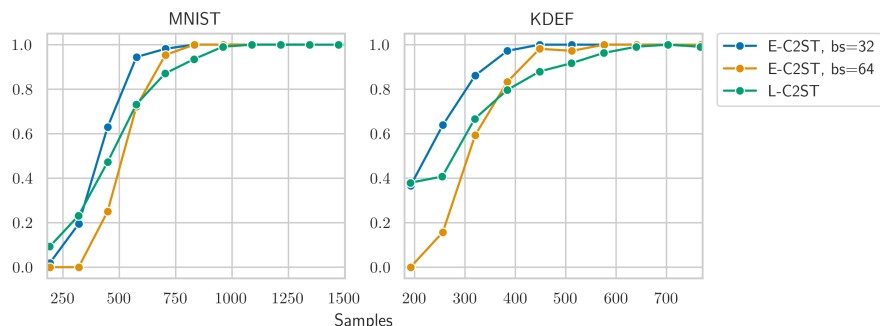

Figure 7: Comparison between the E-C2ST trained with batch size=32 and the initial E-C2ST and baseline L-C2ST. We observe a significant increase in power when using batch size of 32.

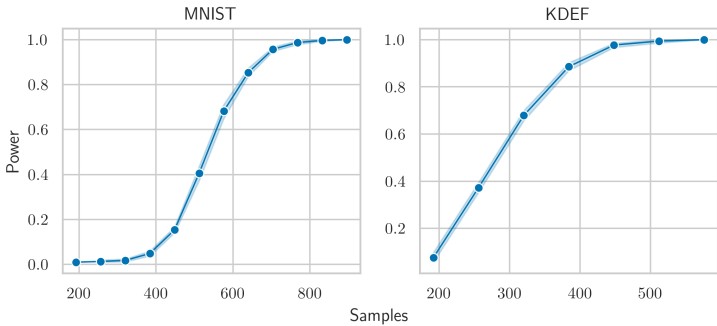

Figure 8: Robustness to batch order. Here we visualize the mean and 95% confidence interval based on 10 batch permutations of the power curves computed from 100 independent runs.

comparing the best E-C2ST performer according to the ablation studies with the best baseline L-C2ST in Figure 7. For both MNIST and KDEF, we can conclude that there is a significant gain in performance by using the enhanced E-C2ST.

**Robustness to the batch order.** We aim to empirically assess the robustness of our statistical test to changes in the order of data batches. To achieve this, we design a sequential experiment in which we iteratively collect new data batches, stopping the process when the null hypothesis is rejected. This experiment is repeated 100 times, with each iteration involving a random shuffling of the predetermined data batches, repeated 10 times to generate 10 different sequence orders. For each sequence order, we calculate the power of the test based on the results of the 100 independent runs. We then aggregate the results from all 10 sequences by calculating and visualizing the mean power curve and its corresponding 95% confidence interval in Figure 8 for the MNIST and KDEF datasets, where the batch size is 64 and the initial $\lambda = 0.5$. Figure 8 shows that the observed deviations from the mean are small in both scenarios. This indicates that our method is robust with respect to permutation in the order of the data batches.

**Wall clock time.** We performed a comparative analysis of the execution time, measured in seconds, for the E-C2ST and L-C2ST algorithms. This analysis is based on 100 independent trials with different sample sizes within the KDEF and MNIST scenarios, as shown in Table 1. In this context, data were generated under the alternative hypothesis. Both E-C2ST and L-C2ST were trained with identical training parameters (learning rate, patience, network architectures). Batches of 64 samples were used for E-C2ST. For L-C2ST, the total sample size - shown in the second row of the Table 1 - was divided according to a ratio we used in all our experiments, i.e. 5:1:1 for training, validation and test data, respectively.

We do not to include the other baselines, since their execution times are comparable to those of L-C2ST. This similarity in execution time is due to the fact that all baselines, including L-C2ST, use the same trained

model and the same number of permutations to compute the corresponding $p$-values. All experimental runs were performed on NVIDIA GP102 [GeForce GTX 1080 Ti] GPUs.

| | KDEF | | | MNIST | | | | |
|---|---|---|---|---|---|---|---|---|
| | $64 \cdot 3$ | $64 \cdot 6$ | $64 \cdot 9$ | $64 \cdot 3$ | $64 \cdot 6$ | $64 \cdot 10$ | $64 \cdot 14$ | $64 \cdot 17$ |
| E-C2ST | $\mathbf{1.5 \pm 0.6}$ | $\mathbf{4.4 \pm 1.4}$ | $\mathbf{5.6 \pm 2.7}$ | $\mathbf{2.5 \pm 0.4}$ | $4.9 \pm 0.8$ | $10.5 \pm 1.8$ | $\mathbf{11.5 \pm 2.8}$ | $\mathbf{11.6 \pm 2.9}$ |
| L-C2ST | $2.7 \pm 1.1$ | $4.6 \pm 2.1$ | $9.4 \pm 3.4$ | $3.5 \pm 0.7$ | $\mathbf{4.4 \pm 0.98}$ | $\mathbf{7 \pm 2.3}$ | $11.6 \pm 3.99$ | $15.96 \pm 5.1$ |

Table 1: Wall Clock Times in Seconds

Our discussion of the results is organized around three different scenarios: 1) when the number of data batches equals three; 2) when the total sample size falls below the threshold required to achieve maximum statistical power; and 3) when the total number of samples exceeds the minimum data set size required to achieve maximum power.

Looking at the first columns of the Table 1 for both the KDEF and MNIST scenarios, we see that E-C2ST requires less computation time compared to L-C2ST when the dataset is segmented into three batches. In this case E-C2ST performs a single training iteration (as L-C2ST) utilizing the three batches for training, validation, and testing, respectively. Then, the reason for the increased computational cost for L-C2ST can be attributed to permutation test used for calculating the $p$-value that uses 500 permutations.

In our work, as demonstrated by our experimental results, including those shown in Figure 8, we have empirically determined the optimal sample sizes for maximizing the statistical power of E-C2ST when using a batch size of 64. In particular, we found that approximately 384 samples (equivalent to $64 \times 6$) are required for KDEF and 896 samples (equivalent to $64 \times 14$) are required for MNIST. This indicates that to effectively reject the null hypothesis, E-C2ST requires a maximum of 6 and 14 batches in the KDEF and MNIST scenarios, respectively.

When the available data set is smaller than these thresholds, E-C2ST becomes more computationally expensive than L-C2ST (see the second and third columns for the MNIST scenario). This increased computational cost occurs because E-C2ST updates the model with each additional batch of data, using the full dataset available at that time, whereas L-C2ST trains the model only once using the train, validation, test data split.

However, when the dataset size exceeds the identified thresholds for achieving maximum power with E-C2ST, E-C2ST becomes less computationally expensive than L-C2ST (see the second and third columns for KDEF and the fourth and fifth columns for MNIST). This efficiency gain is due to E-C2ST's ability to terminate further training upon early rejection of the null hypothesis in large datasets. This means that for datasets larger than the specified thresholds, the computational time for E-C2ST becomes constant and does not increase with additional data, providing a significant computational advantage over L-C2ST.

**Why can't we use $p$-values for sequential testing?** Traditional statistical tests, such as the t-test or chi-squared test, assume that the number of observations or experiments is determined before data collection begins. This setup means that researchers decide in advance how much data to collect, and that predetermined sample size does not change no matter what the outcome of the test is. Sequential testing, on the other hand, allows data to be analyzed as it comes in. This approach allows researchers to make decisions about whether to continue collecting data at any point during the study. Therefore, sequential testing focuses on designing tests that can keep the type I error rate below a certain significance level throughout the data collection period. This is a much more strict requirement than in classical statistical testing, where controlling for type I error is only necessary for a single, fixed sample size.

Standard testing methods are not directly applicable to sequential testing scenarios. In situations where data arrives in batches over time, recalculating the $p$-value for the entire dataset each time new data arrives and making testing decisions based on that recalculated value is likely to push the type I error rate above the alpha threshold. This increased risk is due to the fact that performing multiple tests on the same data increases the likelihood of incorrectly rejecting the null hypothesis. We illustrate this in the following example.

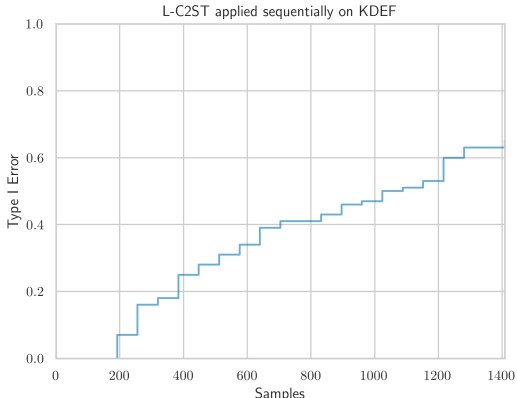

Figure 9: Type I error of L-C2ST, applied on a sequential testing task. We fix the number of batches at 20 and compute the type I error over time, as the number of batches revealed to the model increases.

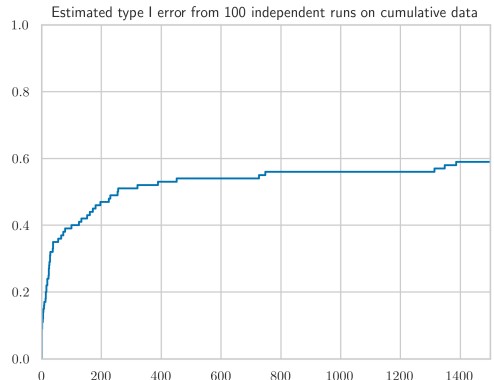

Figure 10: Estimated type I error from 100 independent sequential experiments for batch sizes randomly selected in the interval $[32, 64]$. The estimated cumulative type I error is way above the significance level $\alpha = 0.05$ and naturally increases over time.

We ran a sequential testing experiment, where in each round of this experiment, two samples were drawn, with their sizes randomly chosen from the range $[32, 64]$. Both were sampled from a standard normal distribution.

As new batches of data arrived, a t-test was performed to compare the means of the samples collected up to that point (this includes the current batch and the past data). The test was stopped if the null hypothesis was rejected at a significance level of 0.05; if not, new samples were drawn until the null hypothesis was rejected at that level.

This procedure was repeated 100 times to estimate the type I error rate over time. Our experiment shows that the type I error increases over time and reaches a level of 60%, as shown in Figure 10. Let's look at the implications for the tests we are considering in this work. If we use the same batch splitting technique for any standard method as we do for E-C2ST, we compromise the type I error guarantees that would be in place if we calculated the $p$-value only once. In particular, if we have $M$ batches and thus recalculate a $p$-value $M$ times, the standard methods face $M$ times more opportunities to falsely reject the null hypothesis. This significantly increases the likelihood of inflating the type I error rate when evaluated over the entire data set. For example, Figure 9 shows the increasing type I error of L-C2ST as the number of batches increases, where the maximum batch number is 20 and the batch size is 64. Thus, for M=20, in this sequential setting, the type I error of L-C2ST is above $0.6 >> 0.05$.

On the other hand, the way E-C2ST is designed addresses this problem, i.e., it ensures that the type I error in this case remains below the predetermined significance level. See Corollary 3.2 for these theoretical guarantees.

| Layer (type) | Output Shape |
|---|---|
| Linear-1 | [batch size, 30] |
| LayerNorm-2 | [batch size, 30] |
| ReLU-3 | [batch size, 30] |
| Linear-4 | [batch size, 30] |
| LayerNorm-5 | [batch size, 30] |
| ReLU-6 | [batch size, 30] |
| Linear-7 | [batch size, 2] |

Table 2: The network architecture employed in the Blob experiment for all methods .

| Layer (type) | Parameters |
|---|---|
| Conv2d-1 | 16, kernel size=(3, 3), stride=(2, 2), padding=(1, 1) |
| LeakyReLU-2 | negative slope=0.2 |
| GroupNorm-3 | eps=1e-05 |
| Conv2d-4 | 32, kernel size=(3, 3), stride=(2, 2), padding=(1, 1) |
| LeakyReLU-5 | negative slope=0.2 |
| GroupNorm-6 | eps=1e-05 |
| Conv2d-7 | 64, kernel size=(3, 3), stride=(2, 2), padding=(1, 1) |
| LeakyReLU-8 | negative slope=0.2 |
| GroupNorm-9 | eps=1e-05 |
| Conv2d-10 | 1, kernel size=(3, 3), stride=(2, 2), padding=(1, 1) |

Table 3: The network architecture employed in the MNIST experiment for E-C2ST, L-C2ST, S-C2ST, M-C2ST.

| Layer (type) | Parameters |
|---|---|
| Linear-1 | size=32 |
| LayerNorm-2 | |
| ReLU-3 | |
| Dropout-4 | 0.5 |
| Linear-5 | size=32 |
| LayerNorm-6 | |
| ReLU-7 | |
| Dropout-8 | 0.5 |
| Linear-9 | size=1 |

Table 4: The network architecture employed in the KDEF experiments for all baselines.

