# OpenReview forum: "E-Valuating Classifier Two-Sample Tests"
_TMLR — Accepted by TMLR_

### Review · Reviewer_zK4n · 2023-12-22

**Summary Of Contributions:**

This paper proposes a variation of two-sample tests with deep classifiers based on E-values. To motivate the necessity of the proposed method, authors argue that previous deep classifier-based two-sampled tests suffer from inflated type I error when applied sequentially. To this end, recent works dynamically reject the null hypothesis for different batches, and this is where e-value-based sequential test comes from. Specifically, this work extends Lhéritier and Cazals (2018) by introducing e-values for conditional independence testing. This paper provides some theoretical analyses and empirical observations on small synthetic and real-world datasets to support the proposed method.

**Audience:**

Yes

**Broader Impact Concerns:**

Nothing special.

**Claims And Evidence:**

Yes

**Requested Changes:**

Please address concerns above.

**Strengths And Weaknesses:**

Strength:

+ The thorough theoretical analyses seem to be good for archival purposes for future works.

+ The experimental results demonstrate the effectiveness of the proposed method (unless sample size is small; less than ~600).

Weakness:

- Regarding the statement in page 2 "when applied sequentially, they can lead to inflated type I error. In simpler terms, these methods assume that the sample size is known in advance, which can be a drawback in practice.":
why not just use random sample size per sequence to avoid the known sample size during testing? I am not sure this is a good reason to motivate the following claims.

- According to the paper, Ramdas et al. (2022); Grünwald et al. (2020); Shafer (2019) already discussed e-value-based sequential tests, and it seems like the proposed e-value-based two-sample tests fall into a specific case of them. Also, authors argue that they "extend the work of Lhéritier and Cazals (2018) by introducing e-values for conditional independence testing." So the title looks somewhat too general and/or overclaimed.

- Regarding the statement in page 2 "we assume that both hypotheses (H_0 and H_A) occur with equal probability.":
I think this is a strong assumption. Is there any problem on generalizing this, i.e., making the odds to be arbitrary?

- There are some hyperparameters that the values are not specified, such that the experimental results are not reproducible. For example, what are the initial value of \lambda_1 and the significance level \alpha in the experiments? Also, ablation studies on hyperparameters are mostly missing, except for the batch size.

- The test is sequential, so I wonder if the test result can be different if the order of batches becomes different?

- What is the time/memory complexity of the proposed method and how it can be compared with other works, in terms of big-O notations and/or wall-clock time? Two-sample tests are computationally heavy, so the development at the expense of more costs might not be desirable.


Other issues:

- \theta and \Theta first appeared in section 2.1 are never defined.

- Is there a specific reason to define the alternative hypothesis as H_A, such that you abuse notations? Why not just use H_1, like the conventional literature?

- In Algorithm 1 line 9, E^(1) is never computed; note that line 8 starts from E^(2).

- (I don't expect an answer from the authors regarding this) I am not 100% sure if this work is interesting to TMLR's audiences. I see that there are several works on two-sample tests in ML venues, but it seems TMLR had a difficulty to find reviewers interested in this work.

---

> ### Author Response · Authors · 2024-01-11
> **Thank you for your review!**
>
> We sincerely thank you for your time and expertise. We have carefully implemented all your suggestions in our manuscript.

---

> ### Author Response · Authors · 2024-01-11
> **Statement on page 2**
>
> We will explain this statement in more detail below. For classical statistical testing methods, as well as for classifier two-sample testing (C2ST), it's a fundamental requirement that the total sample size (e.g., both the training and test data in C2ST) be predetermined before data collection begins. Therefore, these methods do not work very well for sequential testing situations where the sample size is not known in advance and is not constant. The main problem with using standard statistical tests in these types of sequential testing scenarios is that they can't reliably control for Type I errors. This means that the probability of falsely rejecting a true null hypothesis could end up being higher than the acceptable threshold or significance level, $\alpha$, noted in our introduction.
>
> To support this point, we performed a sequential testing experiment with random sample sizes, as suggested by the reviewer. In this experiment, two samples are drawn sequentially, each with a sample size randomly chosen from the interval [1,10]. Both samples follow a standard normal distribution. At each time a new batch is received, a t-test is performed to compare the sample means from the cumulative data set available at that time. The procedure and further sampling are stopped if we falsely reject the null hypothesis at a significance level of 0.05; otherwise, sampling continues until the null is rejected at that significance level. This experiment was repeated 100 times to estimate the Type I error rate. Using this testing approach, we observed a maximum Type I error, specifically a false positive rate of 100%, which violates all established safeguards against Type I error. We have included a discussion of this example in the supplementary.
>
> We hope that this response addresses the reviewer's concerns. If additional clarification is needed, we are happy to provide further details.

---

> > ### Comment · Reviewer_zK4n · 2024-02-05
> > **Additional comment**
> >
> > If the experiment is the one in Appendix D.3, I think the range of [1,10] is too small. Especially, I am not sure if this kind of test is meaningful when the batch size is 1. If the batch size used in the original experiment is 64, then [32,128] would be a reasonable choice.

---

> ### Author Response · Authors · 2024-01-11
> **Title**
>
> Our title doesn't claim that we are the first to integrate E-values into sequential testing. Rather, the title is a creative play on words that reflects the name of our test, E-C2ST, for which we use classifier two-sample testing as a reliable approach to generate valid E-values. Our title is intended to capture this unique aspect rather than to claim novelty in the field.

---

> ### Author Response · Authors · 2024-01-11
> **Example in the introduction**
>
> Thank you for your insightful comment regarding the assumption on page 2. We agree that this is indeed a strong assumption and appreciate your suggestion to generalize it by allowing arbitrary odds. We have made the revisions in the updated manuscript.

---

> ### Author Response · Authors · 2024-01-11
> **Ablation Study**
>
> Thank you for your valuable feedback on the hyperparameters in our study. We have set the significance level of $\alpha$ to 0.05 as noted in Section 6.2, and the initial value of $\lambda$ is set to 0.5.  We have included an ablation study for the initial value of $\lambda$ in our revised experimental section. In our results, the initial value of $\lambda$ had no significant impact on the test performance in the KDEV scenario, while in the DCGAN-MNIST case, higher $\lambda$ values slightly increased the test performance. This effect can be attributed to the early stages of testing when the performance of the neural network is not optimal due to limited sample sizes. Combined with a small value of $\lambda$, this results in lower E-values for individual batches, which in turn affects the overall E-value of the test. However, the initial $\lambda$ does not significantly change the number of samples required for maximum power.  If there are other specific hyperparameters that we can address, please let us know. Details of our training procedure (other hyperparameters and network architectures) can be found in Section C (Appendix) of our paper.

---

> ### Author Response · Authors · 2024-01-11
> **Order of batches**
>
> This is indeed an important consideration, and the impact varies depending on whether we're dealing with infinite or finite batch data:
>
>
> Infinite batch data: Here, our consistency analysis provides strong guarantees once the learner satisfies the specified conditions. In this case, we guarantee that the null is rejected with probability 1 under the alternative hypothesis, regardless of the batch order.
>
>
> Finite batch data: In practical scenarios where resources are limited and batch sizes are finite, the order of batches could affect the test results under the alternative hypothesis, especially for small sample sizes. This limitation isn't unique to our method, but is also present in the baseline methods we considered due to the splitting of the train-test data. Different splits can lead to different performance results. To address this issue, we propose to repeatedly permute the data. For each permutation, we will apply either our test or a baseline test, yielding e-values for our method and p-values for the baselines. By aggregating these values, we can derive a more robust e-value or p-value, increasing the robustness of the final decision process. For example, in our case, averaging these E-values could be an effective aggregation strategy.
>
>
> Type I error guarantees: It's important to note that in both scenarios, our method maintains its Type I error guarantees regardless of data size or batch order.

---

> > ### Comment · Reviewer_zK4n · 2024-02-05
> > **Additional comment**
> >
> > > Order of batches
> >
> > Maybe I missed, but it would be good to see some experiments on how the test result is different if the order of batches is different.
> >
> > > Time Complexity
> >
> > I recommend reporting the wall time for all compared methods, that would be useful for future works.

---

> > > ### Author Response · Authors · 2024-02-06
> > > **Order of batches**
> > >
> > > We added new experiments to explore the robustness of E-C2ST to the batch order in Appendix D.3 (see Figure 8). We permuted the batch order 10 times, and based on these sequence orders, we computed the mean and 95% confidence intervals of the power curves for both KDEV and MNIST data. From both experiments, we can conclude that our method is robust to the batch order due to the very small confidence intervals.

---

> > > ### Author Response · Authors · 2024-02-06
> > > **Time Complexity**
> > >
> > > Thank you for the suggestion to include the wall clock times of our method. We have included the results in Appendix D.3 based on the MNIST and KDEV data. Here we want to summarize our results.
> > >
> > > We consider two key messages of this experiment, depending on whether or not the data size exceeds the minimum data size for achieving maximum statistical power. When the data size is smaller than this threshold, E-C2ST tends to be slower compared to other baseline methods. This aligns with what we discussed in our previous response. On the other hand, once the dataset size exceeds this threshold, E-C2ST becomes  more computationally efficient than the baseline methods. This efficiency gain is due to E-C2ST's design to stop training and updating once it has sufficient evidence to reject the null hypothesis. As a result, E-C2ST could potentially stop after examining only a small number of data batches, avoiding the need to process all available data. In comparison, baseline methods use the entire data set to compute test statistics, and are therefore computationally more expensive for large data sizes.

---

> ### Author Response · Authors · 2024-01-11
> **Time Complexity**
>
> In our experiments, we updated the neural network using the entire dataset, which indeed makes our method more computationally expensive compared to conventional c2st. However, it's important to note that our statistical test is flexible and can be adapted to different training methods. We expect that implementating online learning techniques could significantly reduce the computational time, i.e. each batch is used only once for training and the computational time is linear in the number of batches and thus comparable to all c2st based tests. While we didn't explore this in the current work, it is certainly a promising direction for future research. We will discuss this in the revised version of our manuscript.

---

> ### Author Response · Authors · 2024-01-11
> **TMLR audience**
>
> We appreciate your concern about the interest of the TMLR audience in our work. We strongly believe that two-sample testing is highly relevant in the context of modern machine learning. Its applications span various important areas, including simulation-based inference, adversarial robustness testing, and out-of-distribution (OOD) detection, change detection, etc.

---

> ### Author Response · Authors · 2024-01-27
> **Notation**
>
> We chose the notation $H_A$ and $p_A$ over $H_1$ and $p_1$ to avoid potential confusion. Throughout our paper, for example in Section 3.1, we often use '1' to denote the first incoming batch. The adoption of  $H_A$ and $p_A$ helps to clearly distinguish between terms related to the alternative hypothesis from batch-related references.

---

> ### Author Response · Authors · 2024-02-06
> **Updated Experiment**
>
> We have updated the experiments for the suggested batch size range [32,64] and observe similar results as before: increasing Type I error over time and, more importantly, Type I exceeding the chosen significance level $\alpha=0.05$. Please refer to Appendix D.3 and Figure 10. We discussed the reason for the test behavior in more detail in Appendix D.3, which we will summarize here.
>
> Traditional statistical tests, such as the t-test, require researchers to choose the data size before the data collection period begins. This means that the sample size is fixed from the start, regardless of the statistical test results. Sequential testing, however, offers a more flexible approach by analyzing data as it arrives and allowing decisions about whether to continue collecting data to be made at any time point. Therefore, sequential testing focuses on designing tests that ensure that the Type I error remains below a certain level **throughout the entire data collection period**, a much stricter requirement than traditional testing, for which the requirement for type I error control is met at single time points.
>
> When it comes to sequential testing, traditional testing methods are not well suited unless additional posthoc adjustmets are not applied. If we're dealing with data that comes in over time, and we recalculate the p-value each time new data is added, there's a high probability that the error rate will exceed the significance level. This risk comes from the fact that testing the same data multiple times for significance can lead to falsely rejecting the true hypothesis more often than allowed.
>
> How is this related to the testing problems we are considering in our work? Applying the batch splitting technique to traditional methods, as we do with E-C2ST, compromises Type I guarantees. If we split the data into M batches and calculate a new p-value each time, the baselines are M times more likely to falsely reject the null hypothesis. This greatly increases the chance of increasing the Type I error higher that the significance level when looking at the data set as a whole (as shown in Figure 9). E-C2ST, however, is designed to address this issue by ensuring that the Type I error rate remains within acceptable limits even when the data is split in this way.  Please refer to Corollary 3.2 for the theoretical guarantees.

---

### Review · Reviewer_RA3k · 2024-01-16

**Summary Of Contributions:**

The paper presents a classifier-based two-sample test that is more data efficient than prior tests by making use of sequential testing. It explains E-variable-based hypothesis testing, conditional E-variables and hypothesis testing, split-likelihood ratio tests, before expanding on these with conditional predictive independence tests to introduce the E-Valuating Classifier Two-Sample Tests (E-C2ST).

The paper provides convincing experimental validation, comparing convincingly to other literature.

**Audience:**

Yes

**Claims And Evidence:**

Yes

**Requested Changes:**

1. Abstract: mention "E-Valuating Classifier Two-Sample Tests (E-C2ST)" explicitly instead of "E-C2ST" + mention "sequential testing" explicitly in the abstract as well to increase searchability.
2. Mention the chain rule of probability around Lemma 2.2 in §2.2 on Conditional E-Variables. It seems to provide good intuitions for what is going on?
3. Below (5), there might be a typo. There is a $P(...) \to \infty$ --- should it $\to 0$?
4. Please refer to the proofs in the appendix from the main text, so it is easy to jump there in the PDF. In particular, "Proof of (3) being an e-variable" was vital for my understanding.
4. For the ablation (Fig 4.), why not run with batch size $1$? I would very much like to see what the trade-off is. (In Active Learning, the lesson is that one has to train for many more epochs than one thinks with very small datasets.) Could you add a figure to the appendix that includes the best baseline as well to make E-C2ST's additional advantage easier to compare?

5. For future work in the conclusion, could you consider the question of data order? There might be some exciting connections to active learning and active testing, which might further boost the data efficiency of this principled approach.

6. For my understanding: How does this two-sample test compare to other methods in ML/deep learning, which measure the mutual information between samples and their distribution: X ~ P_Z, Z ~ Bernoulli, I[X ; Z] ought to measure something similar and might provide another interesting connection for further research?

**Strengths And Weaknesses:**

### Strengths

1. The paper is exceptionally well-written and flows by itself. (It is one of the best-written papers I have reviewed recently.)
2. The theory is well-explained, and sections §1, §2, §3 provide a good introduction for a reviewer who does not know the area well (that is me).
3. The experiments make sense and are convincing. The ablation addresses precisely what I wanted to ask for after reading the earlier sections.

### Weaknesses

While writing this up, I finally understood that the conditional E-variables are only used via the chain rule for the split likelihood piece. While reading it initially, this did not become clear from the main text.

See below for more fine-grained remarks.

### In Summary

The claims are evidenced, and it will be of great interest to readers of TMLR, given the increased data efficiency. This will be of interest in online settings.

---

> ### Author Response · Authors · 2024-01-27
> **Thank you very much for your positive feedback on our work!**
>
> Thank you very much for your positive feedback on our work! All suggested changes are implemented and all typos are corrected in the revised version of the manuscript. (Questions: 1-4)

---

> > ### Comment · Reviewer_RA3k · 2024-02-04
> > **Thank you**
> >
> > Thank you very much for the update! I'm very happy with the revisions and answers.

---

> ### Author Response · Authors · 2024-01-27
> **Question 5**
>
> Thank you for your suggestion. We indeed see that reducing batch size too strongly can lead to reduced power as a function of seen samples, as we observe in new experiments (specifically, with batch size 8 in the DCGAN-MNIST experiment, please refer to the updated Fig 4). We suspect this effect is driven by training instabilities for smaller batch sizes leading to an initial accumulation of E-values smaller than 1. We expect this trend reversal to hold for smaller batch sizes than 8 as well, but we have not run those experiments due to computational constraints (retraining the model after every batch of size 1 would require about 100K model retrainings). We added experiments with batch sizes 8 and 16 to the paper. Note that the current training procedure is particularly unsuited to training with extremely small batches, as we employ early stopping and the batch size determines both training and validation dataset size.

---

> ### Author Response · Authors · 2024-01-27
> **Question 6**
>
> The assumption that the batches are obtained independently is fundamental to the construction of the E variables. This makes it impossible to manipulate the order of the data. However, active learning could be useful to compose the training data before each model update. For example, we could prune each batch before including it in the training set by actively selecting the most informative samples. With this strategy, we could potentially improve the learning efficiency and effectiveness of the model. We discussed this in the revised paper.

---

> ### Author Response · Authors · 2024-01-27
> **Question 7**
>
> Thanks for your insightful question! Indeed, there is an intriguing connection between the mutual information I(X;Z) and the E-variable, which we now demonstrate in Section C of the Appendix (note that in our paper we use $Y$ instead of $Z$ for the binary variable).
>
> In particular, this result shows that the expected log E-variable $E^{(\leq M)}$ at each step $M$ is, on average, limited by a factor proportional to the mutual information I(X;Z) and the size of the data collected up to time point M, i.e. $ E_{X^{(\leq M)},Z^{(\leq M)}}[\log E^{(\leq M)}]\leq |\mathcal{I}^{(\leq M)}|⋅I(X;Z)$, where $|\mathcal{I}^{(\leq M)}|$ is the number of observations up to step M.
>
> If the null hypothesis is true, then I(X; Z) is 0, and thus the log-E-variable is expected to be non-positive. Otherwise, if the alternative is true, we have an upper bound on the expected log growth of the E-variable, which depends on the mutual information and the sample size. Another interesting point under the alternative is that this bound gets "tighter" as the learning algorithm gets closer to the true distribution, which is evident in the proof of the above statement.

---

### Review · Reviewer_8rXQ · 2024-01-23

**Summary Of Contributions:**

The paper presents a deep classifier two-sample test for high-dimensional data based on e-values. The main idea is developed from the split likelihood ratio tests and predictive independence tests. The proposed method can provide anytime two-sample tests for sequential data at any time point. The paper provides some theoretical analysis to understand the behaviour of the proposed method. The paper also conducts various experiments through some simulation and real data applications to demonstrate the effectiveness of the proposed method.

**Audience:**

Yes

**Claims And Evidence:**

Yes

**Requested Changes:**

+ The writing needs to be modified in order to distinguish the existing works and the new methods and contributions proposed in this work.
+ Clarifications regarding my questions in the Weaknesses section (please refer to the section above).

**Strengths And Weaknesses:**

Strengths:
+ The paper tackles an interesting problem which is to perform two-sample tests for sequential data and to provide the test result at anytime
+ The writing is generally clear (although I still have some comments regarding the writing, please find in the Weaknesses section)
+ The proposed methods are developed from various solid works in the literature. The proposed method seems to be reasonable.
+ Theoretical analysis for the proposed method are included (although I couldn’t check in detail)
+ The proposed method seems to be perform well in the conducted experiments

Weaknesses:
+ I find it sometimes hard to distinguish between the existing works and the new method presented in this paper, especially for researchers that do not work directly in this topic. For example, when introducing the conditional e-variables in Section 2.2, it’s unclear whether this concept was already proposed in (Grünwald et al., 2020; Vovk and Wang, 2021) or was developed in this work. Similarly, the concept of M-Split Likelihood Ratio Test, the predictive conditional independence testing, and other theoretical analysis are not clearly described if they’re new in this work or from existing works. The writing should be modified much clearer to distinguish between existing techniques and the new proposed method.

+ In Section 3, bullets 1 and 2, why in bullet 1, an arbitrary method can be used but then in bullet 2, MLE needs to be used? And normally, how the probability p_\theta is chosen (e.g. Normal distribution or any other distributions)? How can we ensure p_A reflect the true distribution on unseen data?

+ In Section 5, the methodology to perform the hyperparameter is not clear to me. Why do we have the formula in Eq. (11)?

---

> ### Author Response · Authors · 2024-01-27
> **Answer to the question regarding the existing works.**
>
> We sincerely thank you for your time and expertise!
>
> We appreciate your concern about the distinction between our method and pre-existing approaches, and we would like to clarify that we do, in fact, distinguish our work from existing methods in text, particularly in the Contribution part in the Introduction and also at the beginning of Section 4.  As explicitly stated, our work combines the split-likelihood testing procedure of Wasserman et al. (2020) (introduced in Section 3) with the predictive conditional independence framework of Burkart and Király (2017) (introduced in Section 4). This combination within a more general conditional independence framework, especially in the sequential testing setting, is a contribution of our work. Moreover, we also stated in the Introduction that the theoretical analysis on the type I error and the consistency of the proposed E-variables in Sections 3 and 4, are novel contributions. In addtion, we even compare our theoretical analysis in Section 3.1 to an existing related one. For the sake of clarity, we have emphasized all these points more throughout the text.
>
> Regarding the (conditional) E-variables discussed in Section 2.2, these are not a contribution of this paper. To avoid confusion, we made this clear at the beginning of Section 2, whose general purpose is to serve as a background introduction to hypothesis testing with E-variables.

---

> ### Author Response · Authors · 2024-01-27
> **Estimation methods in the M-split-likelihood-ratio framework**
>
> The estimation methods for both null and alternative distributions were specified by Wasserman et al. (2020),  the authors of the M-split likelihood ratio test. In the context of our work, the rationale behind this is that the use of MLE to estimate the distribution under the null hypothesis is crucial because it allows the construction of a valid conditional E-variable, independently of the learning procedure used to estimate the distribution under the alternative. For a more detailed understanding of the intuition behind this E-variable, please refer to Remark 3.1 in the main text.

---

> ### Author Response · Authors · 2024-01-27
> **The choice of hypothesis spaces**
>
> Regarding the choice of $p_{\theta}$, there are no explicit constraints on the $\mathcal{H}_0$ and  $\mathcal{H}_A$ spaces for constructing an E-variable, as described in Section 3. The "design" of these spaces is tailored to the specific testing problem considered. For example, in Section 5, we illustrate how one can construct them for the two-sample testing case. As a reminder in Section 5, we consider the hypothesis $H_0: P(Y)=P(Y∣X)$. Under the null, the Bernoulli random variable $Y$ is independent of $X$. Thus, $\mathcal{H}_0$ includes all Bernoulli distributions (independent of X). As for $\mathcal{H}_A$, in our work we limit it to Bernoulli distributions depending on X that can be parameterized by a neural network. Thus, $\mathcal{H}_A$ represents a broad class of classifiers, but can be further specialized based on the practitioner's interests, such as considering logistic regression models or even models with fixed (non-learnable) parameters. In any of those cases, our methodology guarantees that the resulting E-variable has anytime-valid type I error control.

---

> ### Author Response · Authors · 2024-01-27
> **"How can we ensure p_A reflect the true distribution on unseen data?"**
>
> Under the alternative, ensuring that the distribution $p_A$, parametrized by a neural network,  approximates well the true unknown distribution is a common challenge in the classifier two-sample testing literature. This challenge is closely related to the question of how to choose an optimal network architecture for the statistical testing task at hand.  This remains an interesting direction for future work, e.g. one potential solution is to employ AutoML techniques. Fortunately, our theoretical results (Theorem 3.3 and Theorem 3.4) suggest, that it's not important for $p_A$ to exactly match the true distribution in order to correctly reject the null. Rather, it is more important that $p_A$ approximates the true distribution more accurately than the null model does which is a much milder requirement.

---

> ### Author Response · Authors · 2024-01-27
> **Hyperparameter estimation**
>
> The constrained optimisation problem for the hyperparameter is motivated by Lemma 5.1 (as noted in the main text), which tells us that the special choice of $\tilde{p}_A$, which is a convex combination of the distribution under the null and the learned distribution under the alternative, satisfies the requirements of Theorems 3.3 and 3.4 for a consistent test. However, while Lemma 5.1 provides the theoretical motivation, it does not specify the optimal method for choosing $\lambda_m$.
>
> To fill this gap, we propose a practical solution: estimate $\lambda_m$ that maximizes the conditional E variable from the previous step, i.e., $\lambda_m \in \arg\max_{\lambda\in(0,1)}\log E^{(m-1)}$. This can be interpreted as $\lambda_m$ being part of the learnable parameters of the training procedure for $p_A$. The key point here is that $\lambda_m$ is optimized to find a balance in our beliefs about the true hypothesis - determining at each step $m$ whether the distribution under the null hypothesis or the trained model under the alternative hypothesis better fits the data.

---

> ### Author Response · Authors · 2024-02-01
> **Claims And Evidence**
>
> We noticed that the reviewer selected the "No" option for claims and evidence, indicating that our claims may not have been sufficiently supported with clear evidence. Could the reviewer please specify which specific claims were found to be unsupported?

---

### Author Response · Authors · 2024-01-27
**Thank you for your feedback!**

We are grateful to the reviewers for their constructive feedback. Their insights have been invaluable in improving our paper. We have carefully implemented all their suggestions in the updated version, which now includes additional experiments exploring the effect of the initial value of the mixing parameter $\lambda$, as well as experiments for the ablation study with batch sizes 8 and 16.

We would also like to bring to your attention a revision in Theorem 3.4.  In the original version, when formalizing the conditions on the learner, we inadvertently used the  KL divergence $KL(P_{\theta}\|P_0^{|x^{(M)}})$ instead of the correct expression $ E_{\theta}[\log\frac{p_\theta(x^{(M)})}{p_0(x^{(M)}|x^{(M)})}]$, which resembles the initial expression. We want to assure you that this change does not alter the theorem's consistency result, its proof, or its interpretation. We apologise for this oversight and for any confusion caused by this error.

---

### Decision · Action_Editor_LsbK · 2024-03-29

**Recommendation:** Accept as is

**Comment:**

The submission appears to be of good quality and reviewers only had minor comments.

Reviewer RA3k requests: For the camera-ready, I'd only request one more proof-read to e.g. fix: "from (Burkart and Király, 2017) with e-variables from the M-split likelihood ratio test from Section 3 based on (Wasserman et al., 2020)" to use \citet instead of \citep.

**Audience:**

Classifier based 2-sample testing has a clear precedent in the machine learning literature.

**Claims And Evidence:**

The submission claims a novel sequential classifier-based 2-sample test based on E-variables (non-negative variables bounded by one).  Soundness (non-asymptotic type 1 error control), consistency in sequential and non-sequential settings, milder consistency conditions than previous literature in the 2-sample setting.  Empirical results demonstrate earlier convergence to maximum power than competing methods while maintaining soundness.

Reviewers were unanimous in their recommendation of acceptance of the submission.